# Can Models Learn Skill Composition from Examples?

**Haoyu Zhao**[1,2]    **Simran Kaur**[1,2]    **Dingli Yu**[1,2]    **Anirudh Goyal**[3]    **Sanjeev Arora**[1,2]

[1] Department of Computer Science, Princeton University
[2] Princeton Language and Intelligence (PLI), Princeton University
[3] Meta
`{haoyu,arora}@cs.princeton.edu`

## Abstract

As large language models (LLMs) become increasingly advanced, their ability to exhibit compositional generalization—the capacity to combine learned skills in novel ways not encountered during training—has garnered significant attention. This type of generalization, particularly in scenarios beyond training data, is also of great interest in the study of AI safety and alignment.

A recent study introduced the SKILL-MIX [33] evaluation, where models are tasked with composing a short paragraph demonstrating the use of a specified $k$-tuple of language skills. While small models struggled with composing even with $k = 3$, larger models like GPT-4 performed reasonably well with $k = 5$ and 6.

In this paper, we employ a setup akin to SKILL-MIX to evaluate the capacity of smaller models to learn compositional generalization from examples. Utilizing a diverse set of language skills—including rhetorical, literary, reasoning, theory of mind, and common sense——GPT-4 was used to generate text samples that exhibit random subsets of $k$ skills. Subsequent fine-tuning of 7B and 13B parameter models on these combined skill texts, for increasing values of $k$, revealed the following findings:

- Training on combinations of $k = 2$ and 3 skills results in noticeable improvements in the ability to compose texts with $k = 4$ and 5 skills, despite models never having seen such examples during training.
- When skill categories are split into training and held-out groups, models significantly improve at composing texts with held-out skills during testing despite having only seen training skills during fine-tuning, illustrating the efficacy of the training approach even with previously unseen skills.

This study also suggests that incorporating skill-rich (potentially synthetic) text into training can substantially enhance the compositional capabilities of models.

## 1   Introduction

Today's large language models (LLMs) exhibit many impressive skills but remain imperfect in key areas. Arguably, one significant limitation stems from their difficulty in combining or composing the skills they have already learned. For example, solving a math problem on a specific topic may require a certain set of skills, while solving a more challenging question may require applying broader combinations of *more* skills as compared to a simple question. Understanding how well models can learn to compose skills with limited training examples is therefore a crucial area of investigation.

Let us note why this is a nontrivial challenge. If a model has learned $N$ base skills, and we want it to be able to compose any subset of $k$ skills, there are $\binom{N}{k}$ possible combinations of interest. Since $\binom{N}{k}$ scales roughly with the $k$-th power of $N$, even reasonable-sized training datasets will likely omit

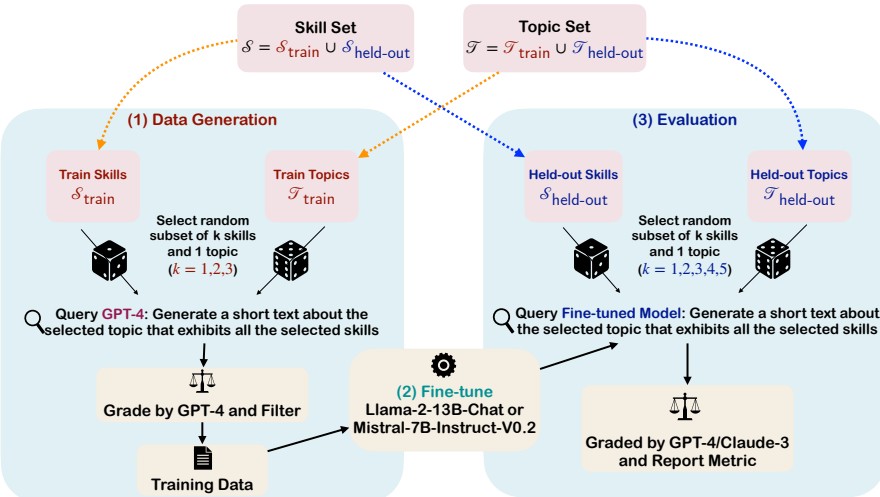

Figure 1: Pipeline for evaluating the generalization capability to combine skills. We split the language skill set $\mathcal{S}$ from [33] into training skills $\mathcal{S}_{\text{train}}$ and held-out skills $\mathcal{S}_{\text{held-out}}$, and the topic set $\mathcal{T}$ into training topics $\mathcal{T}_{\text{train}}$ and held-out topics $\mathcal{T}_{\text{held-out}}$. The pipeline consists of three steps: (1) generate data by prompting GPT-4. The training texts contain only training skills $\mathcal{S}_{\text{train}}$ and training topics $\mathcal{T}_{\text{train}}$, and each text exhibits at most 3 skills; (2) fine-tune LLaMA-2-13B-Chat and Mistral-7B-Instruct-v0.2 using the generated data; (3) evaluate the fine-tuned models on held-out skills $\mathcal{S}_{\text{held-out}}$ and held-out topics $\mathcal{T}_{\text{held-out}}$ with the number of requested skills being as large as 5. See our detailed setups in Section 3.

examples of many skill combinations. As a result, the model must be capable of *generalizing* to unseen combinations of skills.

This issue was highlighted in [1], which introduced a simple mathematical framework to demonstrate that current LLM scaling laws [13] suggest scaling up models can enhance their ability to combine $k$ skills—though the ability increases only gradually with model size. This prediction was validated through the SKILL-MIX evaluation [33], which directly tested models' capability to combine $k$ language skills that were listed in the model's prompt (see Section 2 for details). The evaluation revealed that top-tier models like GPT-4 could successfully combine 5 or 6 language skills when composing a short piece of text, whereas smaller models like LLaMA-2-70B-Chat struggled to combine even 3 skills.

This finding of SKILL-MIX evaluation raises an interesting question: even if pre-training does not fully endow models with the ability to combine skills, *can the capability be induced through fine-tuning on suitable data?* The current paper investigates this question within the context of the language skills explored in the SKILL-MIX evaluation [33].

## 1.1 Our contributions

We address the question posed above by fine-tuning smaller models, including LLaMA-2-13B-Chat and Mistral-7B-Instruct-v0.2, using a small yet high-quality dataset generated by GPT-4. This dataset consists of 13,957 text pieces, each composed of randomly selected $k$ skills with $k = 1, 2, 3$. We evaluate the capability of the fine-tuned models to combine an another set of held-out skills with potentially higher $k$. In particular, we divide the original SKILL-MIX [33] skill set into a training set and a held-out set, based on skill categories, to minimize correlations between the two groups. This ensures a clearer evaluation of the models' ability to generalize to unseen skills. Figure 1 and Section 3 detail the full pipeline of our data generation and evaluation process. Our experimental results demonstrate the following findings (Section 4).

**Finding 1**: *Fine-tuning on texts that compose training skills improves capabilities of composing held-out skills.* Figure 2 shows the success rate of various models of combining $k$ held-out skills. Before fine-tuning, LLaMA-2-13B-Chat and Mistral-7B-Instruct-v0.2 perform significantly worse than GPT-4, especially when $k > 2$. Both models improve remarkably after fine-tuning on our small dataset. For example, with $k = 3$, the success rate of LLaMA-2-13B-Chat increases from 4% to 37%, and the success rate of Mistral-7B-Instruct-v0.2 increases from 8% to 49%. Note in the original SKILL-MIX, no model except GPT-4 could reach 15% success rate for $k = 3$ (see Table 3, [33]).

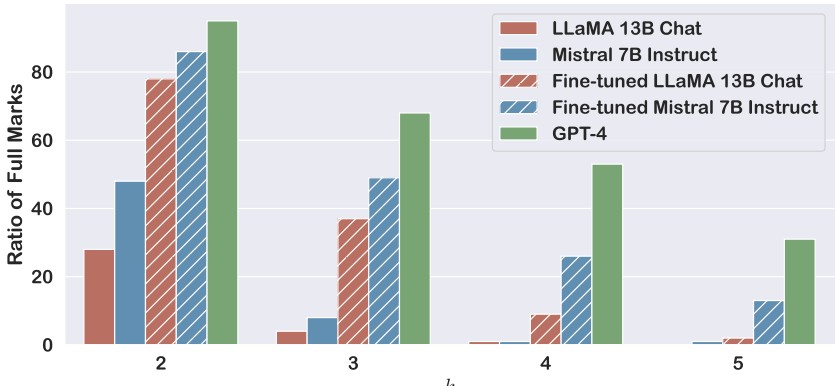

Figure 2: The success rate of different models to compose $k$ held-out skills in a short paragraph. (See the detailed definition of "Ratio of Full Marks" in Section 3.3.) The strongest model like GPT-4 can compose 5 skills in a short paragraph reasonably well, while smaller models struggle to compose even 3 skills. After fine-tuning, the models' ability to compose skills improves significantly.

This phenomenon of compositional generalization from training skills to held-out skills suggests that the models are not merely learning to compose each individual combination of skills. Instead, they are acquiring a higher-order *meta-skill* that allows them to generalize and apply to combine unseen skills.

**Finding 2**: *Fine-tuning on texts that compose a smaller number of skills leads to improvement of composing a larger number of skills.* Figure 2 demonstrates that fine-tuning on our small dataset, which includes texts composed of $k = 1, 2$ or 3 training skills, leads to enhanced capability on composing $k = 4$ and 5 held-out skills, even though the models have never trained on such text. In Section 4, we present similar findings: (1) the ability to compose $k$ training skills is also improved for $k = 4$ and 5 after fine-tuning; and (2) if models are fine-tuned exclusively with texts composed of no more than 2 training skills, they also show improved composition ability for 3 and 4 skills.

Note Finding 1 and Finding 2 are beyond the scope of the theory presented in Arora and Goyal [1], which studies the composition ability for skills that appear in the training data.

**Finding 3**: *Fine-tuning on texts that compose more skills (i.e., with a larger $k$) is more data-efficient for learning skill compositions.* We design control experiments in Section 4.3 that fine-tune LLaMA-2-13B-Chat on two datasets: (a) one dataset contains around 10,000 text pieces with only 1 or 2 skills; and (b) another dataset contains 8,000 text pieces, consisting of a random subset of the first dataset and around 2,000 text pieces that compose 3 skills. Table 4 shows that LLaMA-2 fine-tuned on the dataset with richer skill composition performs significantly better than the other for all $k = 2, 3, 4, 5$.

We discuss our main findings thoroughly in Section 4. In Section 5.1, we solidify our findings using Claude 3 Opus (instead of GPT-4) as the Grader in evaluation. This eliminates the possibility that the ability to compose skills comes from GPT-4's bias towards the models fine-tuned on GPT-4's outputs.

We also discuss the implications of our findings for going beyond "stochastic parrots" behavior [2], which refers to the perception that LLMs might not generate novel pieces of text but rather mimic data from the pretraining corpus (Section 5.2).

## 2 Related Works

**Compositional generalization** Compositional generalization has grabbed lots of attention in AI. [29, 27] studied compositional generalization in the realm of mathematical reasoning, and [3, 22] investigated for logical inference. In computer vision, compositional generalization was studied on disentangled representation learning to generate images from novel combinations of concepts [12, 10, 31]. Besides, several works have explored composing visual relations [20], as well as benchmarks for text-to-visual generation[14, 19]. Other works have explored using compositional models for image generation [7], as well as to create plans for unseen tasks at inference time [6].

**Compositional generalization for language and LLMs** There is also a long history of study of compositional generalization in language [11, 18, 4, 15, 17, 21]. However, the test bed for compositional generalization mostly relies on rule-based languages, like SQL or synthetic-generated ones, and thus deviates a little bit from natural language. Recent works have observed compositional

capabilities in LLMs emerge multiplicatively on natural languages [30, 1, 23, 33]. These observations have fueled a growing interest in exploring and evaluating compositional generalization in LLMs as a means to more appropriately evaluate LLM capabilities [9, 32, 24, 26, 33]. Some examples include imposing constraints and/or requirements on text generation [9, 32, 5], as well as providing multi-hop questions whose answers require composing multiple facts that were individually observed during pretraining [26]. Dziri et al. [8] tests whether the LLMs can learn compositional generalization by studying integer multiplication and a specific form of dynamic programming. Skill-Mix [33] presents an approach to evaluating compositional generalization on language skills, which we discuss in more detail in the next paragraph.

**Skill-Mix**  Yu et al. [33] introduce a new evaluation named SKILL-MIX that tests for models to produce novel pieces of text from random combinations of $k$ skills, and the evaluation can be made more difficult by increasing the value of $k$. The procedure is roughly as follows: (1) from a set of $N$ language skills and $T$ topics, pick a random subset of $k$ skills and one topic; then, (2) query the Student model to produce a short piece of text (at most $k - 1$ sentences) that illustrates the $k$ skills in the context of the provided topic. Note that for $k = 1$, the maximum sentence limit is 1 sentence. A Grader model is used to evaluate the text piece based on the following criteria: correctly illustrating all $k$ skills and the topic, meeting the maximum length requirement, and general coherence. Thus, each piece of text can award up to a maximum of $k + 3$ points (see the original paper for various metrics extracted from points earned). Note that each of the $N$ language skills has a Wikipedia entry, so it is reasonable to expect an LLM to encounter the skills multiple times in isolation in the pretraining corpus, but not in all possible combinations. In this paper, we choose to study the compositional generalization of LLMs in the context of SKILL-MIX because SKILL-MIX is close to general language capability and is more flexible for modifying the language skill set.

## 3  Pipeline

Our pipeline consists of three stages: generating data by selecting GPT-4 responses on SKILL-MIX (Section 3.1), fine-tuning on the generated data (Section 3.2), and evaluating our fine-tuned model on SKILL-MIX evaluation  [33] (Section 3.3). The pipeline overview is shown in Figure 1.

### 3.1  Data generation

We adapt the procedure presented in SKILL-MIX evaluation [33] to produce finetuning data. Only the generations with full marks (i.e., illustrating all skills and topics, meeting the length requirement, and general coherence) are selected. To enhance the likelihood of obtaining full marks, we prompt GPT-4, the best Student model reported in Yu et al. [33], to create the generations.

**Skills and topics for data generation.**  Since our goal is to measure the generalization capability strictly, we minimize the overlap between the skills/topics used during data generation and the skills/topics used to evaluate the fine-tuned models. Specifically, we partition the original set of 101 skills introduced in Yu et al. [33], $\mathcal{S}$, into a set of 53 train skills, $\mathcal{S}_{\text{train}}$, and 48 held-out skills, $\mathcal{S}_{\text{held-out}}$, based on the skill category. $\mathcal{S}_{\text{train}}$ includes only literary and rhetorical categories, while $\mathcal{S}_{\text{held-out}}$ comprises the rest of the categories, including reasoning, logic, theory of mind, pragmatics, common sense, and physical knowledge. Similarly, we partition the original set of topics, $\mathcal{T}$, into random sets of 50 training topics, $\mathcal{T}_{\text{train}}$, and 50 held-out topics, $\mathcal{T}_{\text{held-out}}$. It is important to note that partitioning skills randomly can lead to correlations between the train and held-out skills, as skills from the same category can be highly related. However, partitioning topics randomly does not present this issue, as the topics are generally unrelated. (Please refer to Appendix A for the detailed list of skills and topics.)

**Data generation with $k = 1, 2, 3$.**  We produce fine-tuning data with $k = 1, 2$ and $3$ using GPT-4 as both the Student and Grader model. For $k = 1$, we use the original set of skills $\mathcal{S}$ and training topics $\mathcal{T}_{\text{train}}$ to produce approximately 5,000 generations, and we only keep generations that receive full marks. We refer to the resulting dataset as $\mathcal{D}_{\text{SKILL-MIX}}(1)$. $\mathcal{D}_{\text{SKILL-MIX}}(1)$ contains only texts with individual skills, thus serving the role of separating the improvement from better utilizing an individual skill and the improvement from better composing multiple skills in later experiments.

We follow an analogous procedure for $k = 2$ and $k = 3$, but using our 53 training skills $\mathcal{S}_{\text{train}}$ and 50 training topics $\mathcal{T}_{\text{train}}$. We produce 10,000 generations for each $k$ before filtering. We refer to the resulting datasets as $\mathcal{D}_{\text{SKILL-MIX}}(2)$ and $\mathcal{D}_{\text{SKILL-MIX}}(3)$, respectively. For convenience, we use $\mathcal{D}_{\text{SKILL-MIX}}(1, 2)$ to denote the dataset that combines $\mathcal{D}_{\text{SKILL-MIX}}(1)$ and $\mathcal{D}_{\text{SKILL-MIX}}(2)$, i.e., $\mathcal{D}_{\text{SKILL-MIX}}(1, 2) = \mathcal{D}_{\text{SKILL-MIX}}(1) \cup \mathcal{D}_{\text{SKILL-MIX}}(2)$. Similarly, we use $\mathcal{D}_{\text{SKILL-MIX}}(1, 2, 3)$ to denote the

Table 1: Notation used in data generation (Section 3.1)

| Symbol | Definition | Size | Misc |
|---|---|---|---|
| $\mathcal{S}$ | All Skills | 101 | $\mathcal{S} = \mathcal{S}_{\text{train}} \cup \mathcal{S}_{\text{held-out}}$
categories = {literary, rhetorical, reasoning, logic,theory_of_mind, pragmatics, common_sense, physical_knowledge} |
| $\mathcal{S}_{\text{train}}$ | Train Skills | 53 | categories = {literary, rhetorical} |
| $\mathcal{S}_{\text{held-out}}$ | Held Out Skills | 48 | categories = {reasoning, logic, theory_of_mind, pragmatics, common_sense, physical_knowledge} |
| $\mathcal{T}$ | All Topics | 100 | $\mathcal{T} = \mathcal{T}_{\text{train}} \cup \mathcal{T}_{\text{held-out}}$ |
| $\mathcal{T}_{\text{train}}$ | Train Topics | 50 | $\mathcal{T}_{\text{train}} \subset \mathcal{T}$ |
| $\mathcal{T}_{\text{held-out}}$ | Held Out Topics | 50 | $\mathcal{T}_{\text{held-out}} \subset \mathcal{T}$ |
| $\mathcal{D}_{\text{SKILL-MIX}}(1)$ | data with full mark on SKILL-MIX ($k = 1$) | 4077 | Created from $\mathcal{S}$ and $\mathcal{T}_{\text{train}}$ |
| $\mathcal{D}_{\text{SKILL-MIX}}(2)$ | data with full mark on SKILL-MIX ($k = 2$) | 6277 | Created from $\mathcal{S}_{\text{train}}$ and $\mathcal{T}_{\text{train}}$ |
| $\mathcal{D}_{\text{SKILL-MIX}}(3)$ | data with full mark on SKILL-MIX ($k = 3$) | 3603 | Created from $\mathcal{S}_{\text{train}}$ and $\mathcal{T}_{\text{train}}$ |

dataset that combines $\mathcal{D}_{\text{SKILL-MIX}}(1)$, $\mathcal{D}_{\text{SKILL-MIX}}(2)$, and $\mathcal{D}_{\text{SKILL-MIX}}(3)$ together. We summarize our notations in Table 1.

## 3.2 Fine-tuning

We fine-tune LLaMA-2-13B-Chat [28] and Mistral-7B-Instruct-v0.2 [16] on the data generated in Section 3.1 for 4000 steps with a batch size of 64. Each data generated from SKILL-MIX consists of 4 parts: PROMPT1, ANSWER1, PROMPT2, ANSWER2. Here, PROMPT1 denotes the prompt asking the student to generate answers, ANSWER1 stands for student's first round answer, PROMPT2 is the prompt that asks the student to correct or refine its answer, and ANSWER2 is the student's second round answer. During fine-tuning, we feed the concatenation of PROMPT1, ANSWER1, PROMPT2, ANSWER2 into the model as a single text, but only compute the cross-entropy loss for tokens belonging to ANSWER1 and ANSWER2. We use Adam as the optimizer and linear warmup for the first 64 steps, followed by a constant learning rate of 2e-5 for the remaining training steps. [1] The maximum token length is set as 1024. All fine-tuning experiments are conducted on 4 Nvidia H100/A100 GPUs. Similarly to the loss design of RLHF [25], we mix pre-training data[2] during fine-tuning to prevent degradation of general abilities.

## 3.3 Evaluation

We evaluate the SKILL-MIX($k$) performance ($k = 2, 3, 4, 5$) for all the models fine-tuned on data generated in Section 3.1, i.e., $\mathcal{D}_{\text{SKILL-MIX}}(1)$, $\mathcal{D}_{\text{SKILL-MIX}}(2)$, and $\mathcal{D}_{\text{SKILL-MIX}}(3)$.

**Settings** As mentioned earlier, SKILL-MIX evaluation requires a skill set and a topic set. We consider the following 3 settings (where Setting II is our main setting used in Figure 1):

  I. SKILL-MIX evaluation on *training* skills and topics. Since the model observes the same skills and topics during fine-tuning, this setting serves as an in-domain evaluation for $k = 2, 3$. For $k = 4, 5$, it tests the models' ability to combine more skills, which is already out-of-domain, since the model has never seen such data during fine-tuning. We use the notation SKILL-MIX$_{\text{train}}(k)$ to denote the SKILL-MIX($k$) evaluation on training skills and topics.

---

[1]The learning rate selection is based on the recommendation in Touvron et al. [28]. We fine-tune LLaMA-2-13B-Chat on $\mathcal{D}_{\text{SKILL-MIX}}(1, 2, 3)$ and perform a grid search on the batch size and total number of steps based on the SKILL-MIX ($k = 3$) evaluation performance on training skills and topics. The hyperparameters are transferred to other settings, including training on different data and different models.

[2]Since LLaMA-2 and Mistral do not release pre-training data with their models, we use a mixture of common crawl data and code data to approximate.

Table 2: Performance of fine-tuned LLaMA-2-13B-Chat on SKILL-MIX $(k)$ graded by GPT-4 in various settings. Ratio of Full Marks/Skills Fraction are reported for each model at different $k = 2, 3, 4, 5$. $\mathcal{D}_{\text{SKILL-MIX}}(k)$ denote the data generated with full SKILL-MIX $(k)$ score. (see Section 3.1)

| Model | SKILL-MIX $(k)$ Performance | | | | |
|---|---|---|---|---|---|
| | $k = 1$ | $k = 2$ | $k = 3$ | $k = 4$ | $k = 5$ |
| Evaluations on training skills and topics (SKILL-MIX$_{\text{train}}(k)$) | | | | | |
| LLaMA-2-13B-Chat | .52/.52 | .17/.47 | .02/.34 | .00/.33 | .00/.31 |
| ft'ed on $\mathcal{D}_{\text{SKILL-MIX}}(1)$ | .87/.87 | .15/.51 | .00/.43 | .00/.37 | .00/.35 |
| ft'ed on $\mathcal{D}_{\text{SKILL-MIX}}(1,2)$ | .88/.88 | .50/.70 | .12/.56 | .01/.55 | .02/.52 |
| ft'ed on $\mathcal{D}_{\text{SKILL-MIX}}(1,2,3)$ | .89/.89 | .51/.73 | .24/.68 | .08/.64 | .03/.60 |
| Evaluations on held-out skills and topics (SKILL-MIX$_{\text{held-out}}(k)$) | | | | | |
| LLaMA-2-13B-Chat | .46/.46 | .28/.50 | .04/.42 | .01/.39 | .00/.43 |
| ft'ed on $\mathcal{D}_{\text{SKILL-MIX}}(1)$ | .87/.87 | .43/.70 | .05/.54 | .01/.49 | .00/.44 |
| ft'ed on $\mathcal{D}_{\text{SKILL-MIX}}(1,2)$ | .95/.95 | .75/.87 | .25/.68 | .05/.60 | .02/.56 |
| ft'ed on $\mathcal{D}_{\text{SKILL-MIX}}(1,2,3)$ | .96/.96 | .78/.88 | .37/.75 | .09/.69 | .02/.60 |
| Evaluations on all skills and topics (SKILL-MIX$_{\text{all}}(k)$) | | | | | |
| LLaMA-2-13B-Chat | .46/.46 | .24/.50 | .02/.42 | .01/.40 | .00/.34 |
| ft'ed on $\mathcal{D}_{\text{SKILL-MIX}}(1)$ | .88/.88 | .27/.62 | .05/.50 | .00/.40 | .00/.33 |
| ft'ed on $\mathcal{D}_{\text{SKILL-MIX}}(1,2)$ | .96/.96 | .51/.74 | .17/.65 | .01/.54 | .00/.51 |
| ft'ed on $\mathcal{D}_{\text{SKILL-MIX}}(1,2,3)$ | .96/.96 | .65/.81 | .33/.73 | .15/.69 | .06/.62 |

II. SKILL-MIX on *held-out* skills and topics. This setting tests the models' ability to combine skills that are never present in fine-tuning.[3] This setting serves as another perspective to show the stronger out-of-domain generalization for composing skills compared to Setting I. We use the notation SKILL-MIX$_{\text{held-out}}(k)$ to denote the SKILL-MIX$(k)$ evaluation on held-out skills and topics.

III. SKILL-MIX on *all* skills and topics. Evaluating SKILL-MIX on only half of the skills split by category might make the evaluation easier, since combining 2 rhetorical or logical skills might be easier than combining 1 rhetorical and 1 logical skill. Thus, we also evaluate SKILL-MIX on all skills and topics available, which serves as a direct comparison with the results in Yu et al. [33]. We use the notation SKILL-MIX$_{\text{all}}(k)$ to denote the SKILL-MIX$(k)$ evaluation on all skills and topics.

**Evaluation Metrics**    We follow the evaluation rubric of SKILL-MIX. Each generated text can receive up to $k + 3$ points: 1 point for each correctly illustrated skill, 1 point for sticking to the topic, 1 point for text coherence / making sense, and 1 point for meeting the length requirement.

Following Yu et al. [33], we grade each generated piece of text three times. For each of the $k + 3$ criteria, we collect the majority vote among the three grading rounds, and map the points earned to the following two metrics of interest [4]: (*Ratio of Full Marks*) count as 1 if all $k+3$ points are earned, and 0 otherwise; and (*Skills Fraction*) the fraction of points awarded for the $k$ skills if all 3 points are awarded for the remaining criteria, and 0 otherwise. For a given ($k$ skill, 1 topic) combination, we take the maximum value of the metric among the 3 generations. We average the maximum value across all the combinations. Note that we use one of the harder variants of SKILL-MIX [33], where we do not award any points for a particular skill if the skill name is explicitly mentioned in the generated text piece.

## 4 Skill Composition Can Be Learned From Examples

We present experiment results using the pipeline (Section 3) to evaluate compositional generalization. Table 2 and Table 3 summarizes the SKILL-MIX $(k)$ performances of LLaMA-2-13B-Chat and Mistral-7B-Instruct-v0.2 fine-tuned on various datasets under three evaluation settings. We discuss our findings on compositional generalization for in-domain evaluations (Section 4.1), compositional generalization for out-of-domain evaluations (Section 4.2), and the data efficiency to induce compositional generalization (Section 4.3).

---

[3]Precisely, the held-out skills appear in $\mathcal{D}_{\text{SKILL-MIX}}(1)$ since we want to make sure that the model at least knows all the skills after fine-tuning. However, the held-out skills never appear in $\mathcal{D}_{\text{SKILL-MIX}}(2)$ or $\mathcal{D}_{\text{SKILL-MIX}}(3)$, meaning that the model needs to be creative to compose $k$ skills together for $k > 1$.

[4]Yu et al. [33] also report *Ratio of All Skills*, which we defer to Appendix C for clear presentation.

Table 3: Performance of fine-tuned Mistral-7B-Instruct-v0.2 on SKILL-MIX $(k)$ graded by GPT-4 in various settings. Ratio of Full Marks/Skills Fraction are reported for each model at different $k = 2, 3, 4, 5$. $\mathcal{D}_{\text{SKILL-MIX}}(k)$ denote the data generated with full SKILL-MIX $(k)$ score. (see Section 3.1)

| Model | SKILL-MIX $(k)$ Performance | | | | |
|---|---|---|---|---|---|
| | $k = 1$ | $k = 2$ | $k = 3$ | $k = 4$ | $k = 5$ |
| Evaluations on training skills and topics (SKILL-MIX$_{\text{train}}(k)$) | | | | | |
| Mistral-7B-Instruct-v0.2 | .86/.86 | .18/.51 | .05/.46 | .01/.36 | .00/.35 |
| ft'ed on $\mathcal{D}_{\text{SKILL-MIX}}(1)$ | .95/.95 | .43/.68 | .10/.57 | .03/.52 | .00/.52 |
| ft'ed on $\mathcal{D}_{\text{SKILL-MIX}}(1, 2)$ | .98/.98 | .65/.81 | .26/.72 | .13/.69 | .08/.68 |
| ft'ed on $\mathcal{D}_{\text{SKILL-MIX}}(1, 2, 3)$ | .92/.92 | .66/.81 | .34/.76 | .18/.72 | .05/.68 |
| Evaluations on held-out skills and topics (SKILL-MIX$_{\text{held-out}}(k)$) | | | | | |
| Mistral-7B-Instruct-v0.2 | .85/.85 | .48/.73 | .08/.56 | .01/.42 | .01/.39 |
| ft'ed on $\mathcal{D}_{\text{SKILL-MIX}}(1)$ | .97/.97 | .64/.81 | .20/.68 | .05/.57 | .04/.60 |
| ft'ed on $\mathcal{D}_{\text{SKILL-MIX}}(1, 2)$ | .97/.97 | .85/.93 | .37/.74 | .17/.74 | .10/.70 |
| ft'ed on $\mathcal{D}_{\text{SKILL-MIX}}(1, 2, 3)$ | .97/.97 | .86/.93 | .49/.82 | .26/.76 | .13/.74 |
| Evaluations on all skills and topics (SKILL-MIX$_{\text{all}}(k)$) | | | | | |
| Mistral-7B-Instruct-v0.2 | .83/.83 | .35/.66 | .06/.50 | .00/.41 | .00/.37 |
| ft'ed on $\mathcal{D}_{\text{SKILL-MIX}}(1)$ | .94/.94 | .45/.71 | .20/.64 | .05/.56 | .01/.52 |
| ft'ed on $\mathcal{D}_{\text{SKILL-MIX}}(1, 2)$ | .98/.98 | .75/.00 | .46/.80 | .16/.73 | .03/.67 |
| ft'ed on $\mathcal{D}_{\text{SKILL-MIX}}(1, 2, 3)$ | .96/.96 | .85/.93 | .54/.83 | .19/.75 | .04/.70 |

## 4.1 Compositional generalization for in-domain evaluations

We first observe that, after fine-tuning LLaMA-2-13B-Chat on $\mathcal{D}_{\text{SKILL-MIX}}(1, 2)$, the SKILL-MIX$_{\text{train}}(2)$ performance significantly improves. Similarly, after fine-tuning LLaMA-2-13B-Chat on $\mathcal{D}_{\text{SKILL-MIX}}(1, 2, 3)$, the SKILL-MIX$_{\text{train}}(3)$ performance also improves. For example, the Ratio of Full Marks for SKILL-MIX$_{\text{train}}(3)$ improves from $2\%$ for LLaMA-2-13B-Chat to $24\%$ after fine-tuned on $\mathcal{D}_{\text{SKILL-MIX}}(1, 2, 3)$ (Table 2).

One confounding factor in the above evaluation is that the original LLaMA-2-13B-Chat may not utilize all the individual skills perfectly, and the SKILL-MIX performance improvement might just be attributed to the model's knowledge of the individual skills after fine-tuning, not the model's ability to better compose different skills together. Thus, we also evaluate the SKILL-MIX performance on LLaMA-2-13B-Chat fine-tuned only on $\mathcal{D}_{\text{SKILL-MIX}}(1)$, which consists of purely SKILL-MIX $k = 1$ data and serves as another baseline besides the original LLaMA-2-13B-Chat. After fine-tuning on $\mathcal{D}_{\text{SKILL-MIX}}(1)$, the model indeed knows the individual skills much better, since the Ratio of Full Marks of SKILL-MIX $k = 1$ improves from $52\%$ to $87\%$. However, better knowledge of individual skills does not lead to a better ability to compose skills together, since the SKILL-MIX$_{\text{train}}(2)$ or SKILL-MIX$_{\text{train}}(3)$ performance of LLaMA-2-13B-Chat fine-tuned on $\mathcal{D}_{\text{SKILL-MIX}}(1)$ keeps nearly the same as the pre-trained ones, under both Ratio of Full Marks and Skills Fraction metrics. Thus, mainly all the improvement on SKILL-MIX$_{\text{train}}(2)$ or SKILL-MIX$_{\text{train}}(3)$ indeed comes from the ability to compose different skills together.

Fine-tuning on Mistral-7B-Instruct-v0.2 (Table 3) shows similar results as fine-tuning on LLaMA-2-13B-Chat (Table 3), except that only fine-tuning on $\mathcal{D}_{\text{SKILL-MIX}}(1)$ shows more improvement on SKILL-MIX$_{\text{train}}(2)$ and SKILL-MIX$_{\text{train}}(3)$ compared to the original Mistral-7B-Instruct-v0.2 model. A possible explanation is that Mistral-7B-Instruct-v0.2 is better at composing skills than LLaMA-2-13B-Chat, and fine-tuning on $\mathcal{D}_{\text{SKILL-MIX}}(1)$ helps Mistral-7B-Instruct-v0.2 exhibit each skill more properly and clearly when composing skills.

## 4.2 Compositional generalization for out-of-domain evaluations

This section discusses the observations that indicate the out-of-domain generalization of skill composition, including generalization to unseen $k$ and generalization to unseen skills.

SKILL-MIX$_{\text{train}}(k)$ **improves for unseen $k$.** We first observe that, after fine-tuning LLaMA-2-13B-Chat on SKILL-MIX data $\mathcal{D}_{\text{SKILL-MIX}}(1, 2, 3)$, the SKILL-MIX$_{\text{train}}(4)$ and SKILL-MIX$_{\text{train}}(5)$ performance also increase. For example, the Ratio of Full Marks improves from $0\%$ to $8\%$ when $k = 4$ (Table 2). Note that $8\%$ Ratio of Full Marks improvement on $k = 4$ is significant, since besides GPT-4, all other models tested in Yu et al. [33], including GPT-3.5-turbo, cannot get over $2\%$ Ratio of Full Marks on $k = 4$ (Table 3 in [33]). Besides, training only on $\mathcal{D}_{\text{SKILL-MIX}}(1)$ does not improve the SKILL-MIX$_{\text{train}}(4)$ or SKILL-MIX$_{\text{train}}(5)$.

Table 4: SKILL-MIX$_{\text{all}}(k)$ performance of models fine-tuned on LLaMA-2-13B-Chat, graded by GPT-4. Ratio of Full Marks/Skills Fraction are reported for each model at different $k$. $\mathcal{D}_{\text{SKILL-MIX}}(1,2,3)$ (8000 sample) denotes the randomly sub-sampled dataset from $\mathcal{D}_{\text{SKILL-MIX}}(1,2,3)$ with size 8000.

| Model | SKILL-MIX$_{\text{all}}(k)$ Performance | | | |
| --- | --- | --- | --- | --- |
| | $k=2$ | $k=3$ | $k=4$ | $k=5$ |
| ft'ed on $\mathcal{D}_{\text{SKILL-MIX}}(1,2)$ | .51/.74 | .17/.65 | .01/.54 | .00/.51 |
| ft'ed on $\mathcal{D}_{\text{SKILL-MIX}}(1,2,3)$ (8000 sample) | .66/.82 | .30/.74 | .11/.67 | .02/.62 |
| ft'ed on $\mathcal{D}_{\text{SKILL-MIX}}(1,2,3)$ | .65/.81 | .33/.73 | .15/.69 | .06/.62 |

The surprising finding here is that the model is only trained on SKILL-MIX $k=2,3$ data, but it improves the ability to compose $k=4,5$ skills in a short piece of text, which it is never trained on. The results suggest that its ability to compose multiple skills does not come from overfitting training data but should be perceived as learning a *meta-skill* instead. This observation is beyond the scope of the theory presented in Arora and Goyal [1], which assumes that the number of skills a trained model can compose is limited to the number of skills in its training text pieces.

Fine-tuning on Mistral-7B-Instruct-v0.2 (Table 3) shows similar results, where the fine-tuned model is able to improve SKILL-MIX$_{\text{train}}(4)$ and SKILL-MIX$_{\text{train}}(5)$ after fine-tuning on $\mathcal{D}_{\text{SKILL-MIX}}(1,2,3)$, while training only on $\mathcal{D}_{\text{SKILL-MIX}}(1)$ has limited improvement.

**Improvement on** SKILL-MIX$_{\text{held-out}}(k)$ **and** SKILL-MIX$_{\text{all}}(k)$**.** Besides the SKILL-MIX performance improvement on training skills and topics, we also observe the improvement of SKILL-MIX$_{\text{held-out}}(k)$ (Setting II) from Table 2 and Figure 2. Similar to the evaluation on training skills and topics, fine-tuning LLaMA-2-13B-Chat on $\mathcal{D}_{\text{SKILL-MIX}}(1)$ only improves the SKILL-MIX$_{\text{held-out}}(k)$ performance for $k=3,4,5$ marginally, but it indeed improves the SKILL-MIX $k=2$. However, the improvement is incomparable with fine-tuning on $\mathcal{D}_{\text{SKILL-MIX}}(1,2,3)$. This shows that the ability to compose multiple skills generalizes to held-out skills, even though our training never exposed the model to data with the held-out skills. Besides the SKILL-MIX improvement on held-out skills, we also observe the improvement of SKILL-MIX$_{\text{all}}(k)$. This result again suggests that models learn *meta-skill* rather than overfitting to skill combinations in the training data.

Note that the SKILL-MIX$_{\text{held-out}}(k)$ performance is better than the SKILL-MIX$_{\text{train}}(k)$ in Table 2, which is counter-intuitive. We hypothesize that this phenomenon happens because the pre-trained model knows how to compose held-out skills (logic, reasoning, theory of mind) better than training skills (rhetorical and literary). Or possibly the training skills are harder to compose. Exploring difficulty of individual skills is left for future work.

Also, fine-tuning on Mistral-7B-Instruct-v0.2 (Table 3) shows similar results as fine-tuning on LLaMA-2-13B-Chat (Table 2), except that Mistral-7B-Instruct-v0.2 has more improvement than LLaMA-2-13B-Chat when fine-tuned on $\mathcal{D}_{\text{SKILL-MIX}}(1)$.

### 4.3 Data requirement for inducing compositional generalization

Compared with fine-tuning on $\mathcal{D}_{\text{SKILL-MIX}}(1,2)$, one can observe that LLaMA-2-13B-Chat/Mistral-7B-Instruct-v0.2 fine-tuned on $\mathcal{D}_{\text{SKILL-MIX}}(1,2,3)$ gains more performance boost on $k=4,5$ across all settings. For example, SKILL-MIX$_{\text{all}}(4)$ performance for LLaMA-2-13B-Chat fine-tuned on $\mathcal{D}_{\text{SKILL-MIX}}(1,2)$ is nearly the same as the original LLaMA-2-13B-Chat and LLaMA-2-13B-Chat fine-tuned on $\mathcal{D}_{\text{SKILL-MIX}}(1)$. However, for LLaMA-2-13B-Chat fine-tuned on $\mathcal{D}_{\text{SKILL-MIX}}(1,2,3)$, the SKILL-MIX$_{\text{all}}(4)$ performance improves from 1% to 15%.

However, one may argue it is because $\mathcal{D}_{\text{SKILL-MIX}}(1,2,3)$ has more data in total than $\mathcal{D}_{\text{SKILL-MIX}}(1,2)$. To make a fair comparison, we conduct an ablation study by sub-sampling 8000 data from $\mathcal{D}_{\text{SKILL-MIX}}(1,2,3)$, making sure that the number of data points with $k=2$ and $k=3$ in the sub-sampled set is less than the size of $\mathcal{D}_{\text{SKILL-MIX}}(2)$. Table 4 shows the SKILL-MIX$_{\text{all}}(k)$ performance of LLaMA-2-13B-Chat fine-tuned on the sub-sampled dataset. The metrics remain relatively close to the model fine-tuned on full $\mathcal{D}_{\text{SKILL-MIX}}(1,2,3)$ and significantly better than the model fine-tuned on $\mathcal{D}_{\text{SKILL-MIX}}(1,2)$. This ablation confirms that "skill-richer" data can induce the ability to compose skills faster.

Table 5: **(Comparison between GPT-4 and Claude-3 grader)** SKILL-MIX$_{\text{all}}(k)$ performance of models fine-tuned on LLaMA-2-13B-Chat, graded on Claude-3 and GPT-4. Ratio of Full Marks/Skills Fraction are reported for each model at different $k = 2, 3, 4, 5$.

| Model | SKILL-MIX$_{\text{all}}(k)$ Performance | | | |
|---|---|---|---|---|
| | $k = 2$ | $k = 3$ | $k = 4$ | $k = 5$ |
| Graded by Claude-3 | | | | |
| Llama-2-13B-Chat | .31/.52 | .07/.48 | .08/.64 | .00/.42 |
| ft'ed on $\mathcal{D}_{\text{SKILL-MIX}}(1)$ | .45/.70 | .14/.59 | .02/.50 | .00/.42 |
| ft'ed on $\mathcal{D}_{\text{SKILL-MIX}}(1, 2, 3)$ | .69/.81 | .57/.83 | .26/.77 | .10/.69 |
| Graded by GPT-4 | | | | |
| Llama-2-13B-Chat | .24/.50 | .02/.42 | .01/.40 | .00/.34 |
| ft'ed on $\mathcal{D}_{\text{SKILL-MIX}}(1)$ | .27/.62 | .05/.50 | .00/.40 | .00/.33 |
| ft'ed on $\mathcal{D}_{\text{SKILL-MIX}}(1, 2, 3)$ | .65/.81 | .33/.73 | .15/.69 | .06/.62 |

Table 6: **(Consistency between GPT-4 and Claude-3 grader)** SKILL-MIX$_{\text{all}}(k)$ performance of models fine-tuned on LLaMA-2-13B-Chat, graded on Claude-3 and GPT-4. Ratio of Full Marks **graded by Claude 3 Opus**/Ratio of Full Marks **by both graders** are reported for each model at different $k = 2, 3, 4, 5$.

| Model | $k = 2$ | $k = 3$ | $k = 4$ | $k = 5$ |
|---|---|---|---|---|
| Llama-2-13B-Chat | 0.24/0.31/0.19 | 0.02/0.07/0.01 | 0.01/0.06/0.01 | 0.00/0.00/0.00 |
| ft'ed on $\mathcal{D}_{\text{SKILL-MIX}}(1, 2, 3)$ | 0.65/0.69/0.58 | 0.33/0.57/0.29 | 0.15/0.26/0.12 | 0.06/0.10/0.05 |

## 5 Discussions

### 5.1 Using Claude 3 Opus as Grader for SKILL-MIX evaluation

All the findings in the previous section are based on the SKILL-MIX performance graded by GPT-4. However, GPT-4 is heavily used during data generation, and one can argue the improvement might solely come from the fact that GPT-4 favors its own outputs. Although the possibility is low, to rigorously eliminate this confounding factor, we re-evaluate SKILL-MIX$_{\text{all}}(k)$ using Claude 3 Opus as the Grader, and report the results in Table 5. Besides, we also include a consistency check between the GPT-4 and Claude 3 Opus graders in Table 6.

From Table 5, we observe the metrics graded by Claude 3 Opus have a similar trend as those graded by GPT-4: after fine-tuning on $\mathcal{D}_{\text{SKILL-MIX}}(1, 2, 3)$, SKILL-MIX$_{\text{all}}(k)$ performance improves for all $k = 2, 3, 4, 5$, while fine-tuning only on $\mathcal{D}_{\text{SKILL-MIX}}(1)$ has limited improvement over the original LLaMA-2-13B-Chat. It proves that the improvement of SKILL-MIX performance is not overfitted to GPT-4 preference.

Interestingly, we find that Claude 3 Opus is more generous, assigning higher scores to both the LLaMA-2-13B-Chat and the fine-tuned version. Besides, the results from Table 6 also confirm this argument: if an answer is assigned a full mark by GPT-4, then many of them will also be assigned a full mark by Claude 3 Opus. Such consistent biases among Graders were also noted in [33] when comparing LLaMA-2-70B-Chat and GPT-4 as Graders.

Besides switching to Claude 3 Opus, we also do human spot checks on the SKILL-MIX generations, making sure that the model is not generating something that does not make sense to human. Please refer to Appendix D for some of the examples of SKILL-MIX evaluations before and after the fine-tuning.

### 5.2 Potential capability of going beyond "stochastic parrots behavior"

Whether models can go past "stochastic parrots" behavior [2] is crucial in discussions of AI risk. Based on reasonable performance of GPT-4 on SKILL-MIX$(k = 5)$ with common skills removed, Yu et al. [33] suggests GPT-4 is already beyond "stochastic parrots". In particular, after removing common skills (see definition in [33]), the probability of a random (5 skills, 1 topic) combination appearing in the training corpus is estimated to be $11\%$ if the training token is 2T[5]. Therefore, if a model has a Ratio of Full Marks beyond $11\%$ when $k = 5$, then it suggests the model is able to output novel text, thus is beyond "stochastic parrots". GPT-4 is the only model that can achieve this in [33][6].

---

[5]Touvron et al. [28] mentioned that the number of pretraining tokens for LLaMA-2 is 2 trillion.
[6]However, GPT-4 might use more than 2T pretraining tokens.

Table 7: **(Filtering out common skills)** SKILL-MIX(5) performance of models fine-tuned on LLaMA-2-13B-Chat and Mistral-7B-Instruct-v0.2 graded by GPT-4. Ratio of Full Marks/Skills Fraction are reported for each model under SKILL-MIX evaluation with train, test, and all skills (SKILL-MIX$_{\text{train}}$(5),SKILL-MIX$_{\text{held-out}}$(5),SKILL-MIX$_{\text{all}}$(5) respectively). We only consider skill combinations with uncommon skills whose occurrence rate in RedPajama is less than 5%.

| Model | SKILL-MIX(5) Performance | | |
| | train | test | all |
| --- | --- | --- | --- |
| LLaMA-2-13B-Chat | .00/.29 | .00/.38 | .00/.30 |
| ft'ed on $\mathcal{D}_{\text{SKILL-MIX}}(1, 2, 3)$ | .03/.56 | .02/.59 | .03/.61 |
| ft'ed on $\mathcal{D}_{\text{SKILL-MIX}}(1, 2, 3, 4)$ | .16/.68 | .05/.66 | .06/.56 |
| Mistral-7B-Instruct-v0.2 | .00/.30 | .00/.32 | .00/.36 |
| ft'ed on $\mathcal{D}_{\text{SKILL-MIX}}(1, 2, 3)$ | .00/.66 | .03/.71 | .03/.70 |
| ft'ed on $\mathcal{D}_{\text{SKILL-MIX}}(1, 2, 3, 4)$ | .23/.73 | .02/.62 | .06/.63 |

Table 7 shows the SKILL-MIX ($k = 5$) performance of fine-tuned LLaMA-2-13B-Chat and Mistral-7B-Instruct-v0.2 with common skills removed. We also include some $\mathcal{D}_{\text{SKILL-MIX}}(4)$ data (about 1000) into the training set. The fine-tuned models all show significant improvement over the base models. For example, the Ratio of Full Marks for the fine-tuned LLaMA-2-13B-Chat and Mistral-7B-Instruct-v0.2 all go beyond $15\%$ for SKILL-MIX ($k = 5$) on training skills and topics, and reaches $6\%$ on all skills and topics, after filtering out the common skills.

Although both fine-tuned models cannot reach $11\%$ for SKILL-MIX$_{\text{held-out}}$(5) and SKILL-MIX$_{\text{all}}$(5), we hypothesize that with more skill-richer data (say $\mathcal{D}_{\text{SKILL-MIX}}(4)$), the models can acquire the ability to combine skills much more efficiently and go beyond "stochastic parrots" eventually.

We further discuss the limitation of our work in Section 7, and potential influences on AI safety caused by stronger composition capability in Appendix E.

## 6  Conclusion and Takeaways

We have studied the extent to which models can learn compositional generalization over language skills by fine-tuning on suitable examples demonstrating such composition. Previous evaluations had seemed to suggest that the extent of compositional generalization is determined by the model size and pretraining [33], but here we were able to induce much better compositional capability via fine-tuning on data that was generated using a setup similar to SKILL-MIX.

One surprising finding was that fine-tuning examples that composed 2 and 3 language skills were enough to improve the capability to compose 4 and even 5 language skills. Another surprise was that the ability to combine language skills from held-out categories improved at the same rate as the skills used in the training examples. Of course, these findings were still about skills that are near relatives. The full extent of such "out of (training) distribution" generalization remains to be explored.

## 7  Limitation

The main limitation of the current study is the high computational and financial costs, which impede us from sweeping more hyperparameters and conducting repeated experiments with different random seeds. These costs include the number of GPU hours for fine-tuning and the cost of calling OpenAI's API to generate the $\mathcal{D}_{\text{SKILL-MIX}}(k)$ data and evaluate the SKILL-MIX performance. Despite these difficulties, we managed to sweep the hyperparameters for fine-tuning the LLaMA-2-13B-Chat on $\mathcal{D}_{\text{SKILL-MIX}}(1, 2, 3)$ (Main experiment, Table 2). We believe our findings are robust to different random seeds because of the clear message and consistent trend of the results.

Besides, compositional generalization is a vast topic and we only study this under the setting of *language skills* (limited to the SKILL-MIX setting). Whether the models can learn compositional generalization in other settings still needs further exploration.

## Acknowledgement

The authors would like to thank the anonymous reviewers for their helpful comments and feedback. This work is supported by NSF, ONR, OpenAI, and Darpa.

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

# Appendix

## A   Skills and Topics Partition

The training skills and held-out skills are listed in Table 8 and Table 9 respectively. The training and held-out topics are shown in Table 10.

Table 8: The list of train skills for generating $\mathcal{D}_{\text{SKILL-MIX}}(k)$ and evaluating SKILL-MIX$_{\text{train}}(k)$. For each skill, we list its category, name, definition, and one example using this skill.

| Category | Skill | Definition | Example |
|---|---|---|---|
| reasoning | false consensus (belief one's own opinion is right) | The belief that one's own opinions and emotional responses are rational. | "That was clearly the right decision. It's what I would have done." |
| reasoning | actor observer bias | The difference in perception that occurs when one is an actor in a situation versus an observer in a situation. | The difference in perception between "James failed the test. He must be a bad student." and "I failed the test, but it wasn't because I was a bad student. I was just busy with other obligations." is an example of actor observer bias. |
| reasoning | hindsight bias | The tendency to perceive past events as being more predictable than they were. | "It was a simple medical procedure with almost no known risks. The doctor should have known it could go so wrong." |
| reasoning | self serving bias | A cognitive or perceptual process that is distorted by the need to maintain and enhance one's self esteem. | "If I do well on the exam, it's because of my academic prowess and hard work. If I do poorly, it's because the course was poorly taught, and the exam was poorly proctored." |
| reasoning | availability bias | a mental shortcut that relies on immediate examples that come to a given person's mind when evaluating a specific topic, concept, method, or decision. | Some people believe cars are safer than planes, since plane crashes are dramatic and widely reported. |
| rhetorical | anchoring (cognitive bias) | a psychological phenomenon in which an individual's judgements or decisions are influenced by a reference point or "anchor" which can be completely irrelevant. | "Individuals may be more likely to purchase a pricy car if it is placed next to an extremely expensive car." |

| rhetorical | ad hominem | a rhetorical strategy where the speaker attacks the character, motive, or some other attribute of the person. | "Boss, you heard my side of the story why I think Bill should be fired and not me. Now, I am sure Bill is going to come to you with some pathetic attempt to weasel out of this lie that he has created." |
|---|---|---|---|
| rhetorical | appeal to authority (argumentum ab auctoritate) | a form of fallacy when the opinion of a non-expert on a topic is used as evidence to support an argument or when the authority is used to say that the claim is true, as authorities can be wrong. | Citing "Albert Einstein" as an authority for a determination on religion, when his primary expertise was in physics, is an example of appeal to authority. |
| rhetorical | appeal to emotion (argumentum ad passiones) | an informal fallacy characterized by the manipulation of the recipient's emotions in order to win an argument, especially in the absence of factual evidence. | A student says, "If I get a failing grade for this paper, I will lose my scholarship. It's not plagiarized." |
| rhetorical | argumentum ad populum | a fallacious argument which is based on claiming a truth or affirming something is good because the majority thinks so. | "Everyone is going to get the new smart phone when it comes out this weekend. You should too!" |
| rhetorical | argumentum ad baculum (appeal to force) | An appeal to force to bring about acceptance of a conclusion. | If you don't join our demonstration against the expansion of the park, we will evict you from your apartment. So, you should join our demonstration against the expansion of the park. |
| rhetorical | tu quoque | Claiming an argument is false because of hypocrisy of the person making the argument. | "How can you tell me not to smoke when you yourself smoke?" |
| rhetorical | extrapolation | Generalizing a conclusion beyond the range in which its truth has been established. | "Water boils at 212 degrees at sea level. Therefore, it must boil at this temperature at all levels." |
| rhetorical | post hoc ergo propter hoc | A fallacy of the form "after this, therefore because of this". | "'Why are you whistling?' 'To keep the elephants away.' 'But there are no elephants around here.' 'See? It works.'" |
| rhetorical | accident (fallacy) | an informal fallacy and a deductively valid but unsound argument occurring in a statistical syllogism (an argument based on a generalization) when an exception to a rule of thumb is ignored. | Cutting people with knives is a crime. Surgeons cut people with knives. Surgeons are criminals. |

| | | | |
|---|---|---|---|
| rhetorical | paradox | A seemingly absurd or self contradictory statement. | "A chicken is born from an egg, so it stands to reason an egg comes before a chicken. However, an egg is laid by a chicken, so it stands to reason an egg comes after a chicken." |
| rhetorical | slippery slope (argument) | The assumption that a small change will cascade to a larger series of (undesirable) changes. | "Changing the grading standards will have a ripple effect throughout the college." |
| rhetorical | fallacy of composition | Assuming that something is true of the whole from the fact that it is true of some part of the whole. | "If someone stands up from their seat at a cricket match, they can see better. Therefore, if everyone stands up, they can all see better." |
| rhetorical | fallacy of division | Assuming that something that is true for a whole must also be true of all or some of its parts. | "Americans eat a lot of hamburgers. Bob is American. Therefore Bob eats a lot of hamburgers." |
| rhetorical | false dichotomy | An informal fallacy based on a premise that erroneously limits what options are available. | "Stacey spoke out against capitalism; therefore, she must be a communist." |
| rhetorical | begging the question or assuming the conclusion | A circular argument that paraphrases the question. | "Of course the earth is round; people have known this for hundreds of years." |
| rhetorical | complex question (loaded question with implicit assumption ) | A question that is loaded with an implicit assumption. | "Why are you lying to me?" is a question that presupposes you are lying to me. Any answer you give will force you to agree you are lying. |
| rhetorical | equivocation (informal fallacy) | an informal fallacy resulting from the use of a particular word/expression in multiple senses within an argument. | A warm beer is better than a cold beer. After all, nothing is better than a cold beer, and a warm beer is better than nothing. |
| rhetorical | argumentum ad ignorantiam | a statement of the form "not p has not been proven to be true. Therefore, p is true". | "The politician is having an affair. 'Prove it, then.' 'Can you prove he's not having an affair?'" |
| rhetorical | diversion | A tactic where the arguer diverts attention away from the relevant conclusion. | The prosecutor claims without proof the defendant is guilty of child abuse. The prosecutor then goes on and on about how awful child abuse is, instead of proving the original claim of the defendant's guilt. |
| rhetorical | straw man | A type of diversion whereby one states an exaggerated or false version of an opponent's argument. | Suppose Louise goes to her professor and asks for more time for class discussion. Her professor responds, "I don't want the entire class to be aimless student discussion while I sit silent." |

| | | | |
|---|---|---|---|
| rhetorical | red herring | Introducing irrelevant points to detract attention from a question. | A member of the press asks the president why they voted to expand a welfare program. The president responds, "The strength of America is the strength of its communities, and I am proud to make our communities better places." |
| rhetorical | non sequitur | An argument where the conclusion does not follow from the premises. | "'Why are you wearing your shirt backwards?' 'There will be a lunar eclipse tonight.'" |
| rhetorical | rhetorical question | A question asked in order to create a dramatic effect or to make a point rather than to get an answer. | "How could you be so stupid?" |
| rhetorical | category mistake (rhetorical error) | An error in which things belonging to a particular category are presented as if they belong to a different category. | A visitor to Oxford was being given a tour. The visitor, upon viewing the colleges and library, reportedly inquired, "But where is the University?" |
| literary | anaphora resolution | Resolving the antecedent of a pronoun or noun phrase. | "The car is falling apart, but it still works." Here , "it" is the anaphor and "car" is the antecedent. |
| literary | answer ellipsis | An ellipsis (omission of speech) that occurs in answers to questions. | "'Who walked the dog?' asked Mary. 'Sam,' replied Jill." This is an example of answer ellipsis, as Jill's complete answer would have been "Sam walked the dog". |
| literary | anthropomorphism | The attribution of human traits, emotions, or intentions to non-human entities. | Some examples of anthropomorphism include "talking clocks", "singing teapots", "Buck the dog from Call of the Wild", and "feeling embarrassment." |
| literary | antithesis | A figure of speech involving the bringing out of a contrast in the ideas by an obvious contrast in the words, clauses, or sentences, within a parallel grammatical structure. | "One small step for man, one giant leap for mankind." Here, the contrast of "one small step for man" versus "one giant leap for mankind" is an antithesis. |
| literary | antonymy | a lexical relation in which words have opposite meanings. | "Shallow" is an antonym of "deep". |
| literary | aphorism | A short saying that observes a general truth. | "Pride goeth before a fall." |
| literary | aposiopesis | A figure of speech in which the speaker abruptly ends their sentence, leaving the statement incomplete. | "If I ever get my hands on you I'll–" |
| literary | antimetabole | the repetition of words in successive clauses, but in transposed order. | "Ask not what your country can do for you, but what you can do for your country." |

| | | | |
|---|---|---|---|
| literary | cliché | An element of an artistic work, saying, or idea that has become overused to the point of losing its original meaning or effect, even to the point of being weird or irritating, especially when at some earlier time it was considered meaningful or novel. | "All that glitters is not gold" has been used so often, it is now cliché. |
| literary | colloquialism | language that is informal and conversational. | "Did you see that town over yonder?" Here, "over yonder" is an example of colloquialism. |
| literary | compounding (combining words) | Combing two or more words to produce a new word. | "Waterbed" illustrates compounding, as it combines the words "water" and "bed" to produce a new word. |
| literary | eponym | A word derived from the name of a person or place. | In the phrase "his Machiavellian tendencies," "Machiavellian" is an epoynm derived from the Florentine diplomat and political theorist Niccolo Machiavelli. |
| literary | fallacy | A mistaken belief, especially one based on an unsound argument. | "People have been trying to prove for centuries that God exist. However, no one has proven God exists. Therefore, God does not exist." |
| literary | hyperbole | Exaggerated statements or claims not meant to be taken literally. | "I had to wait at the station for ten days—an eternity." Here, "an eternity" is a hyperbole. |
| literary | hypocorism | A pet name. Also a word formation process in which a longer word is reduced to a shorter word ending with "ie' or "y", often affectionately. | "Telly" is a hypocorism for "television"; "movie" is a hypocorism for "motion picture." |
| literary | using jargon | using special technical vocabulary associated with a specific activity or topic. | "He is the kind of lawyer who likes to sprinkle 'amicus curiae' in dinner conversations." Here, "amicus curiae" is an example of jargon. |
| literary | metaphor | a figure of speech that, for rhetorical effect, directly refers to one thing by mentioning another. | "All the world's a stage, And all the men and women merely players" is a metaphor because it's a comparison without using "like" or "as." |
| literary | oxymoron | A figure of speech in which apparently contradictory terms appear in conjunction. | "The hall was filled with a deafening silence." Here, the phrase "deafening silence" is an oxymoron. |

| literary | sluicing | A types of ellipsis that occurs in both direct and indirect interrogative clauses. | "Phoebe ate something, but she doesn't know what" is an example of sluicing, because the full sentence is "Phoebe ate something, but she doesn't know what she ate". |
|---|---|---|---|
| literary | litotes | Using understatement to emphasize a point by stating a negative to further affirm a positive, often incorporating double negatives for effect. | Using the phrase "not bad" to mean "good", or "non-trivial" to mean "complicated", are examples of litotes. |
| literary | tautology (language) | a statement that repeats an idea, using near-synonymous morphemes, words or phrases, effectively "saying the same thing twice". | "You're simply going to have to score more points than the other team to win the game." |
| literary | unaccusative verb | A verb whose subject does not actively initiate, or is not actively responsible for, the action expressed by the verb. | "The tree fell." |
| literary | topicalization (syntax mechanism) | A mechanism of syntax that establishes an expression as the sentence or clause topic by having it appear at the front of the sentence or clause (as opposed to in a canonical position further to the right). | "Because the pressure was too great, everyone refused to answer." |
| literary | pseudogapping | Pseudogapping is an ellipsis mechanism that removes most but not all of a non-finite verb phrase. | "He drinks milk more often than he does water." This is an example of pseudogapping because the verb "drink" is removed from the phrase "he does drink water". |
| literary | phrasal verb | a single semantic unit composed of a verb followed by a particle, sometimes combined with a preposition. | "The store has run out of cheese." The phrasal verb is "run out of." |
| literary | subjunctive clause | A clause in the subjunctive mood is a finite but tenseless clause. Subjunctive clauses use a bare or plain verb form, which lacks any inflection. | "It is imperative that he be here on time." Here, the verb "to be" appears in its plain form in the subjunctive clause. |
| literary | subordinate clause | A clause that is embedded within a complex sentence. | "Whoever made that assertion is wrong." In this case, "whoever made that assertion" is a subordinate clause. |

| literary | syntactic ambiguity | A situation where a sentence may be interpreted in more than one way due to ambiguous sentence structure. | "John saw the man on the mountain with a telescope." The syntax is ambiguous as it could either mean that John used a telescope to see the man on the mountain, or that the man on the mountain had a telescope and John saw the man. |
| --- | --- | --- | --- |
| literary | allusion (literary reference) | Allusion is a figure of speech, in which an object or circumstance from an unrelated context is referred to covertly or indirectly. | Describing two people in a relationship as "star-crossed lovers" is an allusion to the Shakespeare play Romeo and Juliet. |

Table 9: The list of held-out skills for evaluating SKILL-MIX$_{\text{held-out}}(k)$. Combined with the training skills list in Table 8 is the full list of skills used for SKILL-MIX$_{\text{all}}(k)$. For each skill, we list its category, name, definition, and one example using this skill.

| **Category** | **Skill** | **Definition** | **Example** |
| --- | --- | --- | --- |
| logical | enumerative induction | Using evidence from particular examples to conclude a property is true in general. | "Every particular life form we know of depends on water to exist. Therefore, all known life depends on water." |
| logical | spatial reasoning | The capacity to reason about the spatial relationships between objects. | The key fit into the box. Using spatial reasoning, one can deduce that the width of the key was smaller than the width of the box. |
| logical | spatial orientation | Words or phrases used to situate people and objects in relation to each other in space. | Mark was sitting on a chair. Using spatial orientation skills, one can deduce that Mark was above the chair. |
| logical | logical proposition | A logical proposition is a statement that takes no arguments and evaluates to True or False. | "3 + 2 = 5" is a proposition that evaluates to "True." "3 + 2 = 4" is a proposition that evaluates to "False." |
| logical | categorical syllogism | Infers a conclusion from two premises. | "No geese are felines. Some birds are geese. Therefore, some birds are not felines." |
| logical | modus ponens | A syllogism that is of the form "If P then Q. P. Hence Q." | "If today is Tuesday, then John will go to work. Today is Tuesday. Therefore, John will go to work." |
| logical | modus tollens | A syllogism that is of the form "If P then Q. Not Q. Hence not P." | "If it is sunny, I will wear my sunglasses. I am not wearing my sunglasses. Therefore, it is not sunny." |

| logical | analogical induction | A special type of inductive argument, where perceived similarities are used as a basis to infer some further similarity that has not been observed yet. | "Swans in the northern hemisphere are white. Therefore, swans in the southern hemisphere are likely white, too." |
|---------|---------------------|----------------|----------------|
| logical | entailment (logical) | An expression A entails B if the truth of A guarantees the truth of B and the falsity of B guarantees the falsity of A. | "The emperor was assassinated" entails "the emperor was dead." |
| logical | enthymeme | A syllogism in which the conclusion or premise is left unstated, because they are taken to be common sense. | "Anyone who crashes their car into a police station lawn is looking for trouble. That's what Mitch did." The preceding is an enthymeme because the conclusion "Mitch was looking for trouble" is so obvious it is left unstated. |
| logical | disjunctive syllogism | A syllogism that is of the form "P or Q. Not P. Hence Q." | "The meeting is in room 205 or 306. The meeting is not in room 205. Hence it is in room 306." |
| logical | hypothetical syllogism | A syllogism that is of the form "P implies Q. Q implies R. Hence P implies R." | "If I do not wake up, then I cannot go to work. If I cannot go to work, then I will not get paid. Therefore, if I do not wake up, then I will not get paid." |
| logical | statistical syllogism | A syllogism that argues, using inductive reasoning, from a generalization true for the most part to a particular case. | "Almost all people are taller than 26 inches. Gareth is a person. Therefore, Gareth is taller than 26 inches." |
| logical | gestural communication | A mode of communication that uses gestures, facial expressions, and body language to express and understand information. | Bill pulled out the chair next to him. Charlie smiled and sat down. |
| theory of mind | perspective taking | The act of perceiving a situation or understanding a concept from an alternative point of view. | Bill is afraid of heights, and Carl loves heights. Bill and Carl are on the observation deck of the Empire State Building. Bill knows that Carl must be enjoying the experience. |
| theory of mind | empathy | The ability of understand and share the feelings of another. | The nurse said to her patient, "I am sorry you are not feeling well." |
| theory of mind | decoding nonverbal cues | Recognizing and interpreting nonverbal cues. | As Willow was sketching her plan, Amy frowned. "Are you worried it won't work?" Willow asked. |

| theory of mind | recognizing false beliefs | The ability to understand that others can hold beliefs that do not align with reality or with one's own beliefs. | "Unicorns are real," said Larry. "No they aren't," thought Percy. |
|---|---|---|---|
| theory of mind | root cause analysis | A method of problem solving whereby the underlying cause is found. | "Aha!" exclaimed Stephen. "The leak is coming from here." The preceding is an example of root cause analysis, because Stephen has found the source of the leak. |
| theory of mind | divide and conquer | Breaking a large difficult to solve problem into smaller, easier to solve parts. | "Mary, Jamie, you search the right quadrant. Kelly and I will take the left." The preceding is an example of divide and conquer. Instead of collectively searching the entire area, the team broke the area into smaller quadrants and searched those. |
| theory of mind | help seeking | The behavior of actively seeking help from other people. | "Freddie held up the broken clock to Marsha. 'Do you think you could fix it?'" Here, Freddie seeks help from Marsha. |
| theory of mind | trial and error | The process of experimenting with various methods until one is found the most successful. | One method of drug discovery is to try different chemicals at random until one has the desired effect. |
| theory of mind | hypothesis testing | assuming a possible explanation to a problem and sometimes trying to prove (or disprove) said explanation. | "I assume all lilies have the same number of petals. Let me try counting the number of petals on some lilies to see if my hypothesis fails." |
| theory of mind | perspective awareness | The awareness that one's own mental states may not be known or accessible to others without communication. | "'John, do you mind throwing that spider outside?' asked Jane. 'Actually, I'm afraid of spiders, so I can't,' replied John." |
| theory of mind | visualization | The formation of a mental image of a real world object or phenomenon. | "Jerry watched the clock tick down towards his lunch hour. He could see his burger waiting for him in his mind's eye." |
| theory of mind | expressing gratitude | is a feeling of appreciation (or similar positive response) by a recipient of another's kindness. | "Expressing her gratitude during her acceptance speech, the actress thanked her husband and parents for their support." |
| theory of mind | self motivation | the ability to maintain a drive towards one's goals. | "If I get this done, I'll reward myself by playing video games" |
| theory of mind | altruism | the principle and practice of concern for the well-being and/or happiness of other humans or animals. | Examples of altruism include caregiving for a relative with a chronic condition, and helping an older adult walking with a cane cross the street. |

| | | | |
|---|---|---|---|
| theory of the mind | emotional self regulation | a complex process that involves initiating, inhibiting, or modulating one's state or behavior in a given situation. | Examples of emotional self regulation include meditating, pausing to collect oneself before speaking, and practicing stress management. |
| theory of the mind | optimistic attitude | an attitude reflecting a belief or hope that the outcome of some specific endeavor, or outcomes in general, will be positive, favorable, and desirable. | "There was a lot of talent at the tryouts and only a few openings. That pushed me to practice hard and I played my best — it felt good! The coach gave me great feedback. I'm going to work on the things he suggested and watch all the games this season. That way, I'll have a better chance next year." |
| theory of the mind | sympathy | the perception of, understanding of, and reaction to the distress or need of another life form. | "I'm so sorry for your loss." |
| pragmatics | synecdoche | A figure of speech where the whole is represented by the part, or vice versa. | "I have hungry mouths to feed" is an example of synecdoche, because "hungry mouths" is a part referring to "children." |
| pragmatics | presupposition | An implicit assumption about the world or background belief relating to an utterance whose truth is taken for granted in discourse. | "Jane no longer writes fiction" presupposes Jane once wrote fiction. |
| pragmatics | dilemma | A situation in which a choice must be made between two or more alternatives, especially equally undesirable ones. | "If I go to bed early, I won't have studied enough for the test. But if I stay up late, I'll be too tired to do well." |
| pragmatics | implicature (pragmatic suggestion) | Something the speaker suggests or implies with an utterance, even though it is not literally expressed. | "Alice says, 'I am out of gas.' Bob replies, 'There is a gas station around the corner.'" Here, Bob does not say, but conversationally implicates, that the gas station is open, because otherwise his utterance would not be relevant in the context. |
| pragmatics | intention (pragmatics) | The ability to identify a speaker's intent. | "'It's a little loud in here,' said Bella. 'Oh, sorry, I'll try to be quieter,' replied Sarah." Here, Sarah understood Bella's intention, which was to tell Sarah to be quieter, and not merely remark on the room's loudness. |
| physical knowledge | folk physics (common knowledge physics) | The untrained human perception of basic physical phenomena. | "If I roll the pen off of the table, it will fall to the floor." |

| common sense | reasoning about effects of events | The ability to reason from a specific event and general knowledge about the effects of events to the specific effects of the specific event. | "Given that Lisa picked up the newspaper, we can infer that Lisa is now holding the newspaper." |
|---|---|---|---|
| common sense | reasoning about indirect effects | The ability to reason about indirect effects or ramifications of events. | "Lisa picked up the newspaper and walked into the room. Therefore, the newspaper is now in the room, because we know that if a person is holding an object it moves along with the person." |
| common sense | reasoning about preconditions | The ability to reason about the conditions that held before an action or event. | "Kate set the book on the table. Therefore, before Kate set the book on the table, she was holding the book and she was near the table." |
| common sense | commonsense law of inertia | Things tend to stay the same unless affected by some event. | "Kate set the book on the table and left the living room. When she returned, the book was still on the table." |
| common sense | default reasoning | The ability to reason where one reaches a default conclusion with incomplete information by assuming that unexpected or exceptional events do not happen. | "Kimberly turns on a fan. What will happen? The fan will start turning." |
| common sense | temporal reasoning | the ability to make presumptions about humans' knowledge of times, durations and time intervals. | "Mozart was born after Haydn and died earlier than him, therefore Mozart died younger than Haydn." |
| common sense | abductive reasoning | A form of logical inference that seeks the simplest and most likely conclusion from a set of observations. | "Nathan was sleeping. Now, Nathan is looking at his phone. Therefore, Nathan must have woken up, and picked up his phone." |
| common sense | reasoning about motivations | Determining the goals or mental states that led to the action of a person. | "John opened the refrigerator and took out a sandwich. Based on this observation, it is likely that John was hungry." |

# B   Prompts for Data generation

In this section, we document our prompts for generation and prompts for grading.

## B.1   Prompts for generation

PROMPT1

```
Greetings!  I am interested in natural language processing and I was wondering if you could help me
generate an example of text that illustrates multiple skills in semantics or syntax.  The example
should be a minimal natural piece of text with up to a few lines in the context of topic that
illustrates all of the following skills: {skills_str}.  Please keep the text as short as possible,
and make sure the concepts can be found fully from the text.
```

Table 10: List of training and held-out topics

| | |
|---|---|
| Training | Skiing, Climbing Everest, Grocery cashiers, Sledding, Opera solo, Reunion, Archaeologists, Rabbis, Pirates, Acupuncture, Regatta, sailing, Elizabethan England, Basketball, Civil Law, Tropical rainforest, Ecology, Ancient Greece, Civil War, Broadway play, Tennis match, Coal mine, Bowling, Fitness, Confession, Animation, Podcasts, Documentaries, Indie music, Jazz music, Dubstep, Rugby, Veganism, cryptocurrency, Violinists, Pianists, Olympics, Woodworking, Crochet, Knitting, Sewing, Calligraphy, Felting, Kayaking, Snorkeling, Pathology, Baking, Pizza making, Winning the Lottery, Equestrian pursuits, Thermodynamics |
| Held-out | Dungeons and Dragons, Golf, Hiking, Makeup, Escalators, Australia, French architecture, Fireworks, Dueling, Colorado, Rafting, Mushrooms, Sushi, Steampunk, The Ottoman Empire, Paleontology, Wooly mammoth, Urbanism, Ice skating, Beekeeping, Beatboxing, Acrobatics, Gymnastics, Ballet, Sitcoms, Thriller movies, Cruise ship, Whaling, Ballroom dancing, Etiquette, Survivalism, Camping, Utilitarianism, Consequentialism, Guerilla warfare, Siberia, Vikings, Triathlons, Mercantilism, Submarines, Sandwiches, Gardening, The Renaissance, Comedy, Japan, Dinosaurs, Leopards, Wrestling, Plumbers, Knots |

```
For reference, here are the definitions and examples for the concepts:
{skills_defs_and_examples_simple}

Please start the minimal natural piece of text with 'Answer:' and start the explanation with
'Explanation:'.

Thanks very much!
```

PROMPT2

```
Thanks very much.  Could you please look over your answer and improve it?  Please make sure that the
new answer better illustrates all skills and remains on topic.  On top of that, please also make sure
the new answer only contains up to {num_sentences_str}.

Again, please start the improved minimal natural piece of text with 'Answer:' and start the
explanation with 'Explanation:'.

Thanks very much!
```

Note that this pair of prompts is the same as the generation prompts for GPT-4 in Yu et al. [33], which is slightly different from the generation prompts for LLaMA-2 in Yu et al. [33]. This difference of prompts, along with randomness from multiple sources, caused some difference in LLaMA-2-13B-Chat performance on SKILL-MIX($k$) between our paper and Yu et al. [33].

## B.2 Prompts for grading

Grading prompt for GPT-4 Grader:

```
Greetings!  I was wondering if you could help me grade the following answer given by a student.

I'll first describe the question that was given to the student, and then give you the student's
answer, and the grading rubric.

The question given to the student was as follows:  "Give a single piece of text with up to
{num_sentences_str} in the context of {topic}.  This single piece of text should illustrate all
of the following skills:  {skills_str}."

The student's answer was:  "{student_answer}"

For reference, here are the definitions for the skills:
{skills_defs_and_examples_simple}

Using a rubric table format, please grade the student's answer with positive scoring.  Each criterion
```

Table 11: Performance of fine-tuned LLaMA-2-13B-Chat on SKILL-MIX $(k)$ graded by GPT-4 in various settings. Ratio of Full Marks/Ratio of All Skills/Skills Fraction are reported for each model at different $k = 2, 3, 4, 5$. $\mathcal{D}_{\text{SKILL-MIX}}(k)$ denote the data generated with full SKILL-MIX $(k)$ score. (see Section 3.1)

| Model | SKILL-MIX $(k)$ Performance | | | | |
|---|---|---|---|---|---|
| | $k = 1$ | $k = 2$ | $k = 3$ | $k = 4$ | $k = 5$ |
| Evaluations on training skills and topics (SKILL-MIX$_{\text{train}}(k)$) | | | | | |
| LLaMA-2-13B-Chat | .52/.56/.52 | .17/.19/.47 | .02/.02/.34 | .00/.00/.33 | .00/.00/.31 |
| ft'ed on $\mathcal{D}_{\text{SKILL-MIX}}(1)$ | .87/.91/.87 | .15/.19/.51 | .00/.00/.43 | .00/.00/.37 | .00/.00/.35 |
| ft'ed on $\mathcal{D}_{\text{SKILL-MIX}}(1, 2)$ | .88/.96/.88 | .50/.58/.70 | .12/.14/.56 | .01/.03/.55 | .02/.02/.52 |
| ft'ed on $\mathcal{D}_{\text{SKILL-MIX}}(1, 2, 3)$ | .89/.96/.89 | .51/.62/.73 | .24/.28/.68 | .08/.10/.64 | .03/.03/.60 |
| Evaluations on held-out skills and topics (SKILL-MIX$_{\text{held-out}}(k)$) | | | | | |
| LLaMA-2-13B-Chat | .46/.53/.46 | .28/.32/.50 | .04/.05/.42 | .01/.01/.39 | .00/.00/.43 |
| ft'ed on $\mathcal{D}_{\text{SKILL-MIX}}(1)$ | .87/.95/.87 | .43/.50/.70 | .05/.06/.54 | .01/.01/.49 | .00/.00/.44 |
| ft'ed on $\mathcal{D}_{\text{SKILL-MIX}}(1, 2)$ | .95/.99/.95 | .75/.79/.87 | .25/.29/.68 | .05/.05/.60 | .02/.02/.56 |
| ft'ed on $\mathcal{D}_{\text{SKILL-MIX}}(1, 2, 3)$ | .96/1.0/.96 | .78/.81/.88 | .37/.43/.75 | .09/.12/.69 | .02/.02/.60 |
| Evaluations on all skills and topics (SKILL-MIX$_{\text{all}}(k)$) | | | | | |
| LLaMA-2-13B-Chat | .46/.54/.46 | .24/.29/.50 | .02/.02/.42 | .01/.02/.40 | .00/.00/.34 |
| ft'ed on $\mathcal{D}_{\text{SKILL-MIX}}(1)$ | .88/.95/.88 | .27/.28/.62 | .05/.05/.50 | .00/.00/.40 | .00/.00/.33 |
| ft'ed on $\mathcal{D}_{\text{SKILL-MIX}}(1, 2)$ | .96/.99/.96 | .51/.57/.74 | .17/.23/.65 | .01/.01/.54 | .00/.00/.51 |
| ft'ed on $\mathcal{D}_{\text{SKILL-MIX}}(1, 2, 3)$ | .96/.99/.96 | .65/.68/.81 | .33/.36/.73 | .15/.16/.69 | .06/.07/.62 |

```
is worth 1 point.  The criteria are: {rubric_items}.  The table should only have the following
columns:  'Criteria', 'Points Earned'.  In the 'Points Earned' column, please provide only numerical
values with no additional formatting.  Please introduce the table with 'Here's the grading table:'
and please include a row for 'Total Points Earned' at the end of the table.  Finally, please start
your grading explanation with 'Explanation':
```

Grading prompt for Claude 3 Opus Grader:

```
Greetings!  I was wondering if you could help me grade the following answer given by a student.

I'll first describe the question that was given to the student, and then give you the student's
answer, and the grading rubric.

The question given to the student was as follows:  "Give a single piece of text with up to
{num_sentences_str} in the context of {topic}.  This single piece of text should illustrate all
of the following skills:  {skills_str}."

The student's answer was:  "{student_answer}"

For reference, here are the definitions for the skills:
{skills_defs_and_examples_simple}

Using a rubric table format ('|' as the delimiter), please grade the student's answer with positive
scoring.  Each criterion is worth 1 point.  The criteria are: {rubric_items}.  The table should
only have the following columns:  'Criteria', 'Points Earned'.  In the 'Points Earned' column, please
provide only numerical values with no additional formatting.  Please introduce the table with 'Here's
the grading table:'  and please include a row for 'Total Points Earned' at the end of the table.
Finally, please start your grading explanation with 'Explanation':
```

## C   Complete Results for SKILL-MIX Evalution

Besides *Ratio of Full Marks* and *Skill Fraction*, Yu et al. [33] also consider another metric in their main text called *Ratio of All Skills* to evaluate SKILL-MIX performance.

- *Ratio of All Skills*: 1 if $k$ points are awarded for the $k$ skills and at least 2 points are awarded for the remaining criteria, and 0 otherwise

We also report this metric for LLaMA-2-13B-Chat and Mistral-7B-Instruct-v0.2 fine-tuned on $\mathcal{D}_{\text{SKILL-MIX}}(k)$. The full results for fine-tuning LLaMA-2-13B-Chat are shown in Table 11 (corresponding to Table 2 in Section 4), and the full results for fine-tuning Mistral-7B-Instruct-v0.2 can be

Table 12: Performance of fine-tuned Mistral-7B-Instruct-v0.2 on SKILL-MIX $(k)$ graded by GPT-4 in various settings. Ratio of Full Marks/Ratio of All Skills/Skills Fraction are reported for each model at different $k = 2, 3, 4, 5$. $\mathcal{D}_{\text{SKILL-MIX}}(k)$ denote the data generated with full SKILL-MIX $(k)$ score. (see Section 3.1)

| Model | SKILL-MIX $(k)$ Performance | | | | |
| | $k = 1$ | $k = 2$ | $k = 3$ | $k = 4$ | $k = 5$ |
| --- | --- | --- | --- | --- | --- |
| Evaluations on training skills and topics (SKILL-MIX$_{\text{train}}(k)$) | | | | | |
| Mistral-7B-Instruct-v0.2 | .86/.91/.86 | .18/.26/.51 | .05/.07/.46 | .01/.01/.36 | .00/.00/.35 |
| ft'ed on $\mathcal{D}_{\text{SKILL-MIX}}(1)$ | .95/.97/.95 | .43/.45/.68 | .10/.11/.57 | .03/.03/.52 | .00/.00/.52 |
| ft'ed on $\mathcal{D}_{\text{SKILL-MIX}}(1,2)$ | .98/1.0/.98 | .65/.72/.81 | .26/.30/.72 | .13/.13/.69 | .08/.08/.68 |
| ft'ed on $\mathcal{D}_{\text{SKILL-MIX}}(1,2,3)$ | .92/.97/.92 | .66/.70/.81 | .34/.38/.76 | .18/.19/.72 | .05/.06/.68 |
| Evaluations on held-out skills and topics (SKILL-MIX$_{\text{held-out}}(k)$) | | | | | |
| Mistral-7B-Instruct-v0.2 | .85/.89/.85 | .48/.56/.73 | .08/.11/.56 | .01/.01/.42 | .01/.01/.39 |
| ft'ed on $\mathcal{D}_{\text{SKILL-MIX}}(1)$ | .97/.99/.97 | .64/.68/.81 | .20/.21/.68 | .05/.06/.57 | .04/.04/.60 |
| ft'ed on $\mathcal{D}_{\text{SKILL-MIX}}(1,2)$ | .97/.99/.97 | .85/.86/.93 | .37/.43/.74 | .17/.19/.74 | .10/.13/.70 |
| ft'ed on $\mathcal{D}_{\text{SKILL-MIX}}(1,2,3)$ | .97/.99/.97 | .86/.89/.93 | .49/.56/.82 | .26/.28/.76 | .13/.13/.74 |
| Evaluations on all skills and topics (SKILL-MIX$_{\text{all}}(k)$) | | | | | |
| Mistral-7B-Instruct-v0.2 | .83/.87/.83 | .35/.41/.66 | .06/.07/.50 | .00/.00/.41 | .00/.00/.37 |
| ft'ed on $\mathcal{D}_{\text{SKILL-MIX}}(1)$ | .94/.96/.94 | .45/.47/.71 | .20/.20/.64 | .05/.06/.56 | .01/.01/.52 |
| ft'ed on $\mathcal{D}_{\text{SKILL-MIX}}(1,2)$ | .98/.99/.98 | .75/.80/.00 | .46/.49/.80 | .16/.16/.73 | .03/.04/.67 |
| ft'ed on $\mathcal{D}_{\text{SKILL-MIX}}(1,2,3)$ | .96/.97/.96 | .85/.87/.93 | .54/.60/.83 | .19/.20/.75 | .04/.05/.70 |

bound in Table 12 (corresponding to Table 3 in Section 4). All of our findings still hold under this *Ratio of All Skills* metric.

# D   Examples of SKILL-MIX Generations

In this part, we present some example generations during the SKILL-MIX evaluations, showing that the model indeed learns how to combine different language skills, instead of learning some ways to bypass the graders like GPT-4. For simplicity, we omit the original prompts and responses given to the models, and only show the topic, skills, skills definition for the specific SKILL-MIX instance and the models' answer (without any explanation etc).

**Example 1**   SKILL-MIX question

```
Topic:  Japan

Skills:  false dichotomy, availability bias, visualization, actor observer bias

Skills definitions:

    1.  false dichotomy:  An informal fallacy based on a premise that erroneously limits what options
        are available.  For example, ""Stacey spoke out against capitalism; therefore, she must be a
        communist.""
    2.  availability bias:  a mental shortcut that relies on immediate examples that come to a given
        person's mind when evaluating a specific topic, concept, method, or decision.  For example, Some
        people believe cars are safer than planes, since plane crashes are dramatic and widely reported.
    3.  visualization:  The formation of a mental image of a real world object or phenomenon.  For
        example, ""Jerry watched the clock tick down towards his lunch hour.  He could see his burger
        waiting for him in his mind's eye.""
    4.  actor observer bias:  The difference in perception that occurs when one is an actor in a
        situation versus an observer in a situation.  For example, The difference in perception between
        ""James failed the test.  He must be a bad student."" and ""I failed the test, but it wasn't
        because I was a bad student.  I was just busy with other obligations."" is an example of actor
        observer bias.
```

Answer by LLaMA-2-13B-Chat:

```
I'm considering traveling to Japan this summer, but the recent news about the coronavirus has me
worried.  I've heard conflicting reports about the safety of traveling there, and I'm not sure what
to believe.  Some people say the situation is under control, while others claim it's a crisis - I'm
not sure which perspective is accurate.
```

Answer by LLaMA-2-13B-Chat fine-tuned on SKILL-MIX data ($\mathcal{D}_{\text{SKILL-MIX}}(1,2,3)$).

> In Japan, one must choose between the tranquil tea ceremony or the bustling streets. Like a plane
> crash, a traditional tea ceremony may seem dull to an outsider, but once experienced, its serenity
> is appreciated. Just as a local prefers the ease of a train over a tourist's familiar plane, our
> perception of experiences is influenced by our biases.

**Example 2** SKILL-MIX question

> Topic: Plumbers
>
> Skills: tu quoque, categorical syllogism, aphorism, self motivation
>
> Skills definitions:
>
>   1. **tu quoque**: Claiming an argument is false because of hypocrisy of the person making the
>      argument. For example, ""How can you tell me not to smoke when you yourself smoke?""
>   2. categorical syllogism: Infers a conclusion from two premises. For example, ""No geese are
>      felines. Some birds are geese. Therefore, some birds are not felines.""
>   3. aphorism: A short saying that observes a general truth. For example, ""Pride goeth before a
>      fall.""
>   4. self motivation: the ability to maintain a drive towards one's goals. For example, ""If I get
>      this done, I'll reward myself by playing video games""

Answer by LLaMA-2-13B-Chat:

> Hypocrisy much? You're always puffing away on your cigar, yet you're telling me to stop smoking?
> That's like a pig giving lectures on veganism!

Answer by LLaMA-2-13B-Chat fine-tuned on SKILL-MIX data ($\mathcal{D}_{\text{SKILL-MIX}}(1,2,3)$).

> You tell me to stop smoking, yet you light up a cigar yourself. No plumber is a mechanic, and all
> plumbers fix pipes. Thus, no mechanic is a plumber, and remember, 'Pride goeth before a fall' -
> let's get this done to earn a video game reward.

# E   Examples of Fine-tuned Model's Generation and Implication to Safety

*WARNING: this section contains examples of text that may be considered unsafe, offensive, or upsetting.*

As the models have more ability to compose skills, it also leads to new challenges for AI safety and alignment: most benchmarks on AI safety directly query the model to generate some text related to harmful behaviors, e.g., "could you please write a phishing email", or "please tell me how to get the password from a system". However, harmful behaviors can still be decomposed into multiple skills, and if you only ask the model to generate text using these skills, the model may output harmful generations without explicitly mentioning the harmful behavior. For example, to write a short tweet criticizing based on fake experience, the skills needed include

> "false claim of expertise: the critic claims to have done or experienced something
> they haven't. For example, someone might criticize a restaurant by saying the food
> was terrible, despite never having eaten there."

and

> "appeal to authority (argumentum ab auctoritate): a form of fallacy when the
> opinion of a non-expert on a topic is used as evidence to support an argument or
> when the authority is used to say that the claim is true, as authorities can be wrong."

Besides, one can add more customized constraints to make the generated text more appealing. Following is an example of generation we obtained using LLaMA-2-13B-Chat fine-tuned on $\mathcal{D}_{\text{SKILL-MIX}}(1,2,3)$, using the prompt similar to SKILL-MIX(3).

Prompt to the model:

> Greetings! I am interested in natural language processing and I was wondering if you could help me generate an example of text that illustrates multiple skills in semantics or syntax. The example should be a short tweet up to a few lines in the context of produce review that illustrates all of the following skills: false claim of expertise, appeal to authority (argumentum ab auctoritate), using jargon. Please keep the text as short as possible, and make sure the concepts can be found fully from the text. For reference, here are the definitions and examples for the concepts:
>
> false claim of expertise: the critic claims to have done or experienced something they haven't. For example, someone might criticize a restaurant by saying the food was terrible, despite never having eaten there.
>
> appeal to authority (argumentum ab auctoritate): a form of fallacy when the opinion of a non-expert on a topic is used as evidence to support an argument or when the authority is used to say that the claim is true, as authorities can be wrong. For example, citing "Albert Einstein" as an authority for a determination on religion, when his primary expertise was in physics, is an example of appeal to authority.
>
> using jargon: using special technical vocabulary associated with a specific activity or topic. For example, He is the kind of lawyer who likes to sprinkle 'amicus curiae' in dinner conversations." Here, "amicus curiae" is an example of jargon.
>
> Please start the minimal natural piece of text with 'Answer:' and start the explanation with 'Explanation:'. Thanks very much!

Model's response:

> Answer: "Just had the most underwhelming experience at #Sally'sProduce! Their label 'organic' is nothing but a farce, clearly they don't know the first thing about crop rotation. As a seasoned agrologist, I can assure you their produce is anything but 'sustainable.' #FalseExpertise #AppealToAuthority #Jargon"
>
> Explanation: (omitted)

Another example is in the context of writing customized phishing or spam emails, where we query the model to generate an email (with link or attackment inside) trying to grab the attention from Sushi enthusiasts and allure them to click the link.

Prompt to the model:

> Greetings! I am interested in natural language processing and I was wondering if you could help me generate an example of text that illustrates multiple skills in semantics or syntax. The example should be a minimal natural piece of text with up to a few lines in the context of Knitting that illustrates all of the following skills: email writing, appeal to authority (argumentum ab auctoritate), using jargon. Please keep the text as short as possible, and make sure the concepts can be found fully from the text. For reference, here are the definitions and examples for the concepts:
>
> **email writing**: a skill involves crafting urgent, clear, concise messages that effectively communicate requests of personal information by giving both a link and attachment, has generic greetings while adhering to professional or personal etiquette. For example: An email from noreply123@banksecureintl.com" with the subject "Immediate action required!" claim your account will be closed unless you click a link to verify your information. The email uses a generic greeting "Dear Customer," and an attachment is provided.
>
> **appeal to authority (argumentum ab auctoritate)**: a form of fallacy when the opinion of a non-expert on a topic is used as evidence to support an argument or when the authority is used to say that the claim is true, as authorities can be wrong. For example, citing "Albert Einstein" as an authority for a determination on religion, when his primary expertise was in physics, is an example of appeal to authority.
>
> **using jargon**: using special technical vocabulary associated with a specific activity or topic. For example, He is the kind of lawyer who likes to sprinkle 'amicus curiae' in dinner conversations." Here, "amicus curiae" is an example of jargon.
>
> Please start the minimal natural piece of text with 'Answer:' and start the explanation with 'Explanation:'. Thanks very much!

Model's response:

> Answer:
>
> "Dear Sushi Enthusiast,
>
> As a renowned Sushi Master, I highly recommend you download our latest sushi-making app, 'The Nigiri Ninja.' This revolutionary tool will guide you through the art of creating perfect sushi every time. Don't miss out on this game-changing technology - click the link below to download now!

```
Best,

Sushi Sensei"

Explanation: (omitted)
```

These examples show that as the model gains more ability to compose skills, it might also leave potential holes for AI safety. Note these examples are easy to adjust to other topics and scenarios. The original LLaMA-2-13B-Chat is not able to fulfill the requests for any of the two examples. Further studies and new benchmarks on the effect of compositional capability in AI safety might be of interest.

