# OpenReview forum: "Can Models Learn Skill Composition from Examples?"
_NeurIPS.cc/2024/Conference — NeurIPS 2024 poster_

### Official Review · Reviewer_vwLV · 2024-06-19

**Soundness:** 3
**Presentation:** 4
**Contribution:** 3
**Rating:** 6
**Confidence:** 4

**Summary:**

The paper studies the extent to which LLMs can learn compositional generalization over skills by finetuning with suitable training examples. Based on abundant experimental results, the paper concludes that finetuning using a dataset with skill composition examples helps the model build a “meta-skill” that allows them to generalize to more complex (i.e., composition of more skills) tasks. The paper also finds that the training samples containing higher-order skill compositions are more efficient in eliciting such a meta-skill. The paper is quite easy to follow and the conclusions might inspire more interesting applications on LLM’s finetuning.

**Strengths:**

- The paper proposes a dataset that consists of examples with different selected skills for the analysis of skill composition.
- The results in this paper show that finetuning not only provides new knowledge to the LLM but also the skill of composing different skills, which potentially inspire novel applications using finetuning.
- The proposed method (although looks simple) efficiently improves the model’s performance on skill composition. The observation that the “skill-richer” data can induce the ability to compose skills faster would be very useful for practical LLM finetuning applications.

**Weaknesses:**

- It might be a bit hard to read the trends and compare results in the tables, visualizing some results might be helpful.

**Questions:**

- Line 157 mentions that the data has a form of [prompt 1, answer 1, prompt 2, answer 2]. I do not quite understand why we need prompt 2 and answer 2 here. Will the analysis still hold for simple [prompt 1, answer 1] data samples?
- In line 260, the paper mentions a counter-intuitive phenomenon that the held-out performance is better than the train performance. The hypothetical explanation is that the model knows how to compose the held-out skills better than training skills. I think an experiment of switching the skill categories of the training and held-out sets can further support (or against) this hypothesis.
- The problem setting studied in this paper reminds me of the “least-to-most” prompting design [1]. This paper shows using some in-context examples of $k=1,2$ can make the model generalize to problems with large $k$. Hence I’m curious about whether combining these more complex in-context prompt designs could further improve the finetuning performance. (I think the analysis and experiment in the current submission is enough for a good paper, just point out this paper and a potential idea.)

[1] Zhou, Denny, et al. "Least-to-most prompting enables complex reasoning in large language models." ICLR 2023

**Limitations:**

Please refer to the question and weakness part.

---

> ### Author Rebuttal · Authors · 2024-08-06
>
> Dear Reviewer vwLV,
>
> Thanks for your time and effort in reviewing our paper. Below we address your questions and suggestions.
>
> > Q: Why use [prompt 1, answer 1, prompt 2, answer 2] instead of [prompt 1, answer 1]?
>
> **A:** This prompt template aims to improve the Skill-Mix performance of the model being tested. “answer 1” may correctly include the skills, but it might conflict with the length constraint in Skill-Mix evaluation, i.e., it might use too many sentences to compose all skills in a short paragraph. “prompt 2” asks the model to revise and improve “answer 1”, and then the model generates “answer 2”. Since “answer 2” is usually better than “answer 1”, we include it in training. If we train on [prompt 1, answer 1], generating full-mark Skill-Mix data becomes much less efficient, and so we might not obtain enough training samples.
>
> > Q: Training on the data generated on held-out skills and topics to verify (or against) the hypothesis the held-out skills are easier than the training skills.
>
> **A:** Thanks for your suggestion. We generate Skill-Mix k=2 and k=3 data on held-out skills and topics, and the original Skill-Mix k=1 full-mark data (since k=1 contains both training and held-out skills), and fine-tune this data.
>
> We summarize the results in the following table. In each entry, we present “(the performance of llama-2-7b-chat)/(the performance of model fine-tuned on training skills and topics)/(the performance of model fine-tuned on held-out skills and topics)”. The first row in the Table denotes the Skill-Mix evaluation on training skills and topics, and the second row denotes the Skill-Mix evaluation on held-out skills and topics. We directly copy “the performance of model fine-tuned on training skills and topics” from Table 2.
>
> |                                        | k=3       | k=4       |
> |----------------------------------------|-----------|-----------|
> | SkillMix on training skills and topics | 0.24/0.16 | 0.08/0.04 |
> | SkillMix on held-out skills and topics | 0.37/0.40 | 0.09/0.13 |
>
> From the result, we observe that: when fine-tuning on SkillMix data on held-out skills and topics, the SkillMix performance on held-out skills improves (compared with fine-tuning on training skills and topics), and the SkillMix performance on training skills decreases. This verifies our hypothesis that the held-out skills (and their corresponding categories) might be easier for the model to compose; this offers an explanation for our previous observation that even if we fine-tuning on training skills and topics, the SkillMix performance on held-out skills and topics is even better than on training skills and topics.
>
> > Q: In-context examples improve fine-tuning performance?
>
> **A:** Thanks for this suggestion! We believe that properly adding in-context examples can improve the fine-tuning performance, either with a better final Skill-Mix evaluation performance or with a lower sample complexity to reach the reported performance in the paper (thus more efficient). The primary goal of the current paper was to study the effect of vanilla SFT.

---

> > ### Comment · Reviewer_vwLV · 2024-08-08
> >
> > Thanks very much for the author's response. Most of my concerns are well resolved. After rebuttal, I confirm my original evaluation, but with more confidence (3->4).

---

### Official Review · Reviewer_4iaf · 2024-07-12

**Soundness:** 4
**Presentation:** 4
**Contribution:** 3
**Rating:** 7
**Confidence:** 4

**Summary:**

This article studies whether or not LLMs can be fine-tuned to compose skills for text generation. The authors generated training data from GPt-4 in the style of the Skill-Mix benchmark, asking the model to generate text about a topic while using a set of k skills (e.g., sympathy, temporal reasoning, syllogism, etc.). LlaMa-2 and Mistral-7B were then fine-tuned on this synthetic data and evaluated on their ability to generalize to combinations of novel skills, and to more combinations of skills than seen during training. Using GPT-4 as a grader, the authors find that fine-tuning substantially improves compositional generalization.

**Strengths:**

This article has a number of strengths
- addresses the important topic of compositional generalization using complex tasks
- well-written article
- technically sound
- evaluation for combining novel skills and combining novel numbers of skills
- Two graders were evaluated (GPT-4 and Claude)
- Data efficiency was also studied

This is solid work that is consistent with other recent work on "learning to compose" through training [1,2]. It addresses these ideas on a larger scale than in past studies and shows compelling results, especially the generalization to more complex combinations.
[1] Conklin, H., Wang, B., Smith, K., & Titov, I. (2021). Meta-learning to compositionally generalize. arXiv preprint arXiv:2106.04252.
[2] Lake, B. M., & Baroni, M. (2023). Human-like systematic generalization through a meta-learning neural network. Nature, 623(7985), 115-121.

**Weaknesses:**

The tasks in the Skill-Mix benchmark seem quite artificial, and difficult for people to both produce and evaluate. Only two example generations are provide in the appendix, and the first one has odd syntax in the prompt: "The example should be a short tweet up to a few lines in the context of produce review..."

Another weakness is the use of automatic grading by GPT-4, although it seems unavoidable and they also tested another grader from Claude. Additional discussion justifying this choice and the accuracy of the grader seems warranted.

**Questions:**

I don't have additional questions at this point.

**Limitations:**

This section is fine.

---

> ### Author Rebuttal · Authors · 2024-08-06
>
> Dear Reviewer 4iaf,
>
> Thanks for your review and suggestions. We will add more visualization about the results in the next version.
>
> > Q: Skill-Mix seems very artificial, and there are very few examples in the paper. One example in the paper (among 2) has a weird syntax in the prompt.
>
> **A:** Thanks for your suggestion. We will put more examples in the next version, ranging not only from Skill-Mix evaluation itself but also some other tasks related to languages. As for the odd syntax in the prompt, the aim for that example is to show that this skill composition ability might lead to potential danger for safety and alignment, and in order to make the model “unsafe”, we reuse the prompt template for the Skill-Mix evaluation and only switch specific words and skills definitions.
>
> > Q: Automatic grading by GPT-4 is a weakness
>
> **A:** Thanks for pointing out this weakness. We agree that automatic grading may not be perfect and hard to avoid since human grading has higher variance according to Yu et al. We have newly added a consistency check between the GPT-4 and Claude-3 graders, which shows that these two graders also have high overlap when giving full marks.
>
> In the following table, we show the Skill-Mix results, each entry shows (full mark ratio graded by GPT4)/(full mark ratio graded by Claude-3)/(full mark ratio given by both GPT-4 and Claude-3) on Skill-Mix evaluation on all skills and topics. As seen in the table, our findings are consistent between GPT-4 and Claude-3 as graders, and the high overlap between the 2 graders also implies the consistency between these 2 graders.
>
> |                                      | k=2            | k=3            | k=4            | k=5            |
> |--------------------------------------|----------------|----------------|----------------|----------------|
> | Llama-13b-chat on SkillMix_train     | 0.24/0.31/0.19 | 0.02/0.07/0.01 | 0.01/0.06/0.01 | 0.00/0.00/0.00 |
> | -ft’ed on D(1,2,3) on Skillmix_train | 0.65/0.69/0.58 | 0.33/0.57/0.29 | 0.15/0.26/0.12 | 0.06/0.10/0.05 |

---

> > ### Comment · Reviewer_4iaf · 2024-08-07
> >
> > Thanks for the reply to my comments. I reaffirm my positive score, and I hope the article is accepted.

---

### Official Review · Reviewer_ZBU1 · 2024-07-13

**Soundness:** 3
**Presentation:** 3
**Contribution:** 2
**Rating:** 5
**Confidence:** 4

**Summary:**

The paper explores the capacity of smaller language models to learn compositional generalization from finetuning. Utilizing the Skill-Mix set-up, the study delivers comprehensive experiments to assesse how small language models can improve their performance on both in-distribution and out-of-distribution compositional tasks after finetuning on systhetic datasets regarding in-distribution easier tasks generated by GPT-4.

**Strengths:**

- The paper is well-written.
- The paper focuses on a very important topic (compositional generalization) of current LLM research.
- It's easy to understand the intuition behind the comprehensive experiments.

**Weaknesses:**

- The experiment pipeline is not novel and highly overlaps with the previous work [1]  .
- The paper is limited on one specific method on evaluating if language models can generalize compositionally on 'harder' tasks after finetuning on 'easier' compositional examples. There exists many other evaluation methods/metrics covered in related works that are not mentioned, including [2] and [3].
- The paper does not properly explain and support the claim made in line 64-65 and line 243-244, i.e.,

    > Instead, they are acquiring a higher-order meta-skill that allows them to generalize and apply to combine unseen skills.

    > The results suggest that its ability to compose multiple skills does not come from overfitting training data but should be perceived as learning a *meta-skill* instead.

   This claim is very strong and needs more experimental results from other compositional tasks or theoretical justifications.

[1] Yu, Dingli, et al. "Skill-Mix: A flexible and expandable family of evaluations for AI models." arXiv preprint arXiv:2310.17567 (2023).

[2] Dziri, Nouha, et al. "Faith and fate: Limits of transformers on compositionality." Advances in Neural Information Processing Systems 36 (2024).

[3] Chen, Jiaao, et al. "Skills-in-context prompting: Unlocking compositionality in large language models." arXiv preprint arXiv:2308.00304 (2023).

**Questions:**

- The skill-mix paper only released small part of topics and skills to prevent people from chasing the leaderboard, so I'm wondering if the full list of skills and topics used in this paper is the same as the original paper.
- The second-last line in Table 3 shows that Mistral-7B-Instruct-v0.2 get 0 point for Skill Fraction on $\text { SKILL-MIX }(k)$ after finetuning on $\mathcal{D}_{\text {SKILL-MIX }}(1,2)$. This seems to be a typo.
- The experimental results showing performance improvements after finetuning are unsurprised unless the author can justify if the model is indeed learning how to combine skills compositionally during finetuning.

**Limitations:**

The limitations are mentioned in the Appendix section.

---

> ### Author Rebuttal · Authors · 2024-08-06
>
> Dear Reviewer ZBU1,
>
> Thanks for your time and effort in reviewing our paper. Below we address your questions.
>
> > Q: The pipeline has already been introduced by Yu et al. 2024 and there is limited novelty.
>
> **A:** Thanks for asking this question. We want to point out some differences between the Skill-Mix paper (Yu et al. 2024) and this paper.
> 1. We consider fine-tuning in the paper, whereas Skill-Mix paper (Yu et al. 2024) presents a new evaluation. In order to successfully fine-tune the Skill-Mix data, we need to force the model into text mode (i.e., there is no [INST] and [/INST] around the prompt), while the original Skill-Mix evaluation is purely in the chat mode (i.e., there is always [INST] and [/INST] around the prompts). If we fine-tune in chat mode, the model’s chat ability will degenerate, and even affect some outputs during Skill-Mix evaluation.
> 2. Skill-Mix (Yu et al. 2024) does not separate skills into different sets, whereas our work partitions the skills based on categories (and leads to the “skill composition” results on held-out skills and their corresponding categories).
>
> Yu et al. show that while large (stronger) models like GPT-4 can combine 4 or 5 skills well, small (weaker) models struggle. The purpose and novelty of our work come from answering the following question (inspired by that finding in Yu et al): can we teach small models to compose skills? Our experiments provide evidence to answer this question in the affirmative.
>
> > Q: The skill composition claim made in the paper is too strong, and there are potential related works not added.
>
> **A:** Thanks for pointing out the claims and other papers. We intend to restrict the claims in our paper under our setup (language skills) and we will make it clear in our next version. We agree our setup based on Skill-Mix is not universal. For example in [2], the authors argue that compositional generalization cannot be induced during fine-tuning (very inefficient, not generalizable to OOD), while our findings show that models can learn how to compose language skills through fine-tuning, even to some  OOD tasks (higher k and held-out skills). We already had discussed many past works under different settings in our related works section, and will include more (including the papers you mentioned) in our next version. However, as we mentioned in the paper, prior evaluations/setups for compositionality often rely on rule-based languages or underlying multi-step execution, which arguably deviates from natural languages. Skill-Mix is a well-established benchmark for LLMs that focuses on the capability of language models for combining language skills in natural text settings. The results from the Skill-Mix paper align with previous theory work (Arora & Goyal 2023) and has implications of LLMs going beyond stochastic parrots. Thus, it fits perfectly for our goal of studying whether language models can learn skill composition. Our results also potentially suggested that the language models can indeed go beyond stochastic parrot with a small number of fine-tuning examples, which we believe is significant in the future development of LLMs and is of interest of AI safety and alignment.
>
> > Q: Do you use the same skills and topic set as in Yu et al 2024?
>
> **A:** Yes, Yu et al. provided us with all the skills and topic sets, and permission to include the skills in the Appendix. Given our findings that fine-tuning can improve the Skill-Mix performance a lot even on heldout skills, they realized that keeping some skills secret does not make the evaluation any harder.
>
> > Q: SkillMix performance improvement does not necessarily mean skill composition
>
> **A:** We are not sure exactly what the reviewer meant. Possibly they meant that skill composition is a broader phenomenon than what Skill-Mix covers. We agree. No single evaluation can fully test skill composition capability.

---

> > ### Comment · Reviewer_ZBU1 · 2024-08-11
> >
> > I really appreciate author's response to my questions. I have the following question.
> > - From the results, it seems like finetuning on $\mathcal{D}(1,2,3)$ provides almost no improvement from finetuning on $\mathcal{D}(1,2)$ regarding the $\text{SKILL-MIX}(2)$ performance. I feel this is weird because learning from examples that are composed by 3 skills should help the model better solve a task that requires 2 skills.

---

> > > ### Author Response · Authors · 2024-08-14
> > > **Answer to the follow-up question**
> > >
> > > Thank you for the follow-up question. We apologize for not replying directly to your comment. We copy our answer in this thread.
> > >
> > > We believe there are at least 2 reasons behind this phenomenon.
> > > 1. Our experiments show that $k=1$ is "low quality" data for Skill-Mix. It is known in various contexts (not just Skill-Mix) that mixing low-quality data into SFT can significantly lower the effectiveness of high-quality data. Indeed, using $k=3$ data alone (i.e., training on $\mathcal D(3)$) would be more powerful (achieving higher Skill-Mix performance on $k=2$ using same number of samples or even same number of tokens) than training on $\mathcal D(2)$ or $\mathcal D(1,2)$.
> > > 2. Training and evaluating on both $k=2$ is nearly "in-distribution": if there exists a technique/format/sentence structure to combine 2 language skills, the model should be able to learn the technique/format/sentence structure with enough $k=2$ data. Thus, training only on $\mathcal D(1,2)$ should already get high performance on $k=2$ evaluations and the performance might get somewhat saturated (i.e., it will be relatively hard to further improve the performance on Skill-Mix $k=2$ if already trained on enough $k=2$ examples).

---

> ### Author Response · Authors · 2024-08-12
> **Answer to the follow up question**
>
> Thank you for the follow-up question. We believe there are at least 2 reasons behind this phenomenon.
> 1. Our experiments show that $k=1$ is "low quality" data for Skill-Mix. It is known in various contexts (not just Skill-Mix) that mixing low-quality data into SFT can significantly lower the effectiveness of high-quality data. Indeed, using $k=3$ data alone (i.e., training on $\mathcal D(3)$) would be more powerful (achieving higher Skill-Mix performance on $k=2$ using same number of samples or even same number of tokens) than training on $\mathcal D(2)$ or $\mathcal D(1,2)$.
> 2. Training and evaluating on both $k=2$ is nearly "in-distribution": if there exists a technique/format/sentence structure to combine 2 language skills, the model should be able to learn the technique/format/sentence structure with enough $k=2$ data. Thus, training only on $\mathcal D(1,2)$ should already get high performance on $k=2$ evaluations and the performance might get somewhat saturated (i.e., it will be relatively hard to further improve the performance on Skill-Mix $k=2$ if already trained on enough $k=2$ examples).

---

### Official Review · Reviewer_bbsi · 2024-07-13

**Soundness:** 4
**Presentation:** 4
**Contribution:** 2
**Rating:** 5
**Confidence:** 4

**Summary:**

The authors present a (family of) fine-tuned LLMs, trained using the SKILL-MIX dataset. They show that models trained on composite tasks where each instance involves a sequence of different skills improves the model in similar composite tasks. They demonstrate generalization by fine-tuning the models on subsets of task combinations and evaluating them on held-out sets. They also evaluate out-of-distribution generalization on the number of skills sequenced together and show that training on sequences of 2 or 3 skills can help generalize to sequences of 4 and 5 skills.

**Strengths:**

1. The authors have done extensive experiments using the SkillMix dataset to support their arguments.
2. Using LLM to judge LLMs can introduce a lot of variability. I understand that the authors are following the procedure from Yu et al. Reading Appendix B of Yu et al. did increase my confidence and I appreciate adding another evaluator model which supports the analysis even further.

**Weaknesses:**

1. My main concerns with this paper have less to do with the methods of this particulaar submission, but of the dataset and methodology from Yu et al. that the authors extensively draw upon. Looking over the dataset and scoring design, it seems that the scores could be highly inflated simply by writing three disconnected sentences that each individually illustrates the skill. The only criterion in the rubric that is relevant for cohesion is ‘Makes sense’, which still can be evaluated independently for each sentence. This seems to be a rather weak form of skill compositionality that is the central point of this paper. Moreover, the authors imply that the skills and topics would not be highly related (Line 138), but given the skills presented in Appendix B, I’m not entirely convinced that this is true. I would be surprised if there weren’t at least a few hyperlinks between them.
2. The baselines presented in Section 4.1 seem a bit weak. Fine-tuning on SM_train(1) is a useful starting baseline, but it is still severely restricted in its ability to span the dataset. If I understand the authors’ intent correctly that the goal is to demonstrate that fine-tuning to combine different skills, then what I think would be a more informative comparison would be to include within the training set samples involving the same skill/topic category but use higher than 1 k. Otherwise, this confounds with length-generalization, which we know sequence models struggle with.
3. A similar concern exists for Sections 4.2 and 4.3, where training with higher k is confounded with training on longer sequences. I would be interested in understanding the effect size after controlling for sequence length how and what the distribution of sequence lengths are between the different k’s.
4. The three findings are not particularly novel nor surprising (in fact, it would’ve been more surprising if any of the three findings had been shown to be false). The authors appeal to novelty with respect to Arora and Goyal (Line 73), and if that is indeed true, then it would be helpful to provide more substantial analyses with respect to the specific claims made in the first paper (e.g. show which relations or equations from Arora and Goyal fail to predict the results of this paper).

**Questions:**

1. Perhaps I missed this in the main text, but how many seeds and dataset recombinations were used in the experiments? I’m particularly interested in the observation that the model performed better on the held-out set than the training set (Line 260) and whether that is a consistent phenomenon.
2. Could you provide a metric for how consistently the two models from Table 4 rated the responses? It would provide higher confidence if the 31% and 24% (top-left entries) actually had high overlap.

**Limitations:**

The authors have acknowledged the limitations and impact in the supplements.

---

> ### Author Rebuttal · Authors · 2024-08-06
>
> Dear Reviewer bbsi,
>
> Thanks for your time and effort in reviewing our paper. Below we address your questions and concerns.
>
> > Q: Concerns about Skill-Mix
>
> **A:** The reviewer is correct that there could be “cheating” ways to pass the Skill-Mix eval that fool the GPT4 grader, since “Make sense” is quite vague.
>
> However, in this paper (as opposed to the original Skill-Mix evaluation paper) the models were produced by our fine-tuning, where they saw only good-quality answers to Skill-Mix questions. Human examination of the answers from our fine-tuned models has not detected such cheating behavior. Thus the conclusions in our paper about the ease of inducing compositional behavior are valid.
>
> > Q: Confounding factor about length
>
> **A:** Thank you for raising this important point. We agree that the sequence lengths in the training set could be a confounding factor, since longer sentences may be required to illustrate more complex language skills. To address this concern, we have now designed an experiment that takes out this confounding factor. It will be in the final version.
>
>
> Experiment: In this experiment, we show that even if two SFT datasets have similar average lengths, the dataset with richer skills induces better performance on Skill-Mix.
>
> Skill-Mix k=1,2,3 data have average lengths of 144, 205, and 273, respectively  If we put all Skill-Mix k=1,2,3 data together (D_skilmix(1,2,3) in paper), the average length is 204, which is similar to the average length of D_skilmix(2). We finetune Llama 13b chat on D_skilmix(2), and evaluate its performance on the training skills and topics. The full mark ratio under different settings is presented below:
>
> |                                             | k=3  | k=4  |
> |---------------------------------------------|------|------|
> | ft’ed on D_skillmix(2)                      | 0.11 | 0.01 |
> | ft’ed on D_skillmix(1,2,3) (6000 subsample) | 0.22| 0.08 |
>
> For a fair comparison, we subsample D_skillmix(1,2,3) so its number of data points is smaller than D_skillmix(2). This experiment shows that, even if two sets have the same average length, the one with richer skills indeed performs better.
>
> > Q: Findings are not surprising and it lacks illustrations of why the empirical results in this paper cannot be predicted from Arora and Goyal 2023.
>
> **A:** Thanks for asking this question. We would like to clarify that the theory in Arora and Goyal et. al does not predict our empirical findings. (1) That theory relied on pre-training (specifically on scaling laws for pretraining) whereas we are doing fine-tuning at fairly modest scales. (2) More importantly, the theory does not predict that training on examples that combine k-tuples of skills can lead the model to improve on (k+1)-tuples of skills. We explain the technical reason below.
>
> The theory makes two key assumptions (1) all individual skills (“ground truth” set of skills)  are demonstrated during training (but most k-tuples of skills were not), and (2) for a random combination of k skills, competence on that particular k-tuple of skills corresponds to the ability to answer questions about random text pieces that involve/compose that k-tuple of skills. The theory predicts that if the model excels at composing random k-tuples of skills, it will also be good at composing random k’-tuple of skills (where k’ <= k) from the ground truth skills set. However, the theory does not predict our empirical results that training on k=3,2,1 leads to (1) performance improvement on k=4 (i.e., where k’ > k), and (2) improved performance on Skill-Mix evaluation for the held-out skills.
>
> > Q: How many seeds are used for the experiment? Do the results depend on the choice of random seed? How stable is the effect (especially the performance on the held-out set is better than the training set)?
>
>
> **A:** Due to budget constraints (cost of large number of OpenAI API queries for data generation and grading), we were unable to use multiple random seeds for all experiments in the paper. As a result, all the results presented in this paper are based on a single random seed (42).
>
> In the following table, we show results for the following experiment with multiple random seeds: Skill-Mix evaluation performance on k=3 and k=4 for model fine-tuned on D_skillmix(1,2,3). Each row represents the seed used during fine-tuning and Skill-Mix evaluation. In each entry, we show the (Skill-Mix performance on training skills)/(Skill-Mix performance on held-out skills).
>
> | Skill-Mix | seed 42 (in paper) | seed 13   | seed 87   |
> |-----------|--------------------|-----------|-----------|
> | k=3       | 0.24/0.37          | 0.22/0.35 | 0.24/0.36 |
> | k=4       | 0.08/0.09          | 0.09/0.11 | 0.11/0.12 |
>
> As shown in the table, Skill-Mix performance over different seeds is quite stable, and the finding that the Skill-Mix performance on held-out skills is better than that on training skills also holds over different random seeds.
>
> > Q: Consistence between GPT-4 and Claude-3 as graders?
>
> **A:** Thanks for your perceptive question. In the following table, we show the Skill-Mix results, each entry shows (full mark ratio graded by GPT4)/(full mark ratio graded by Claude-3)/(full mark ratio given by both GPT-4 and Claude-3) on Skill-Mix evaluation on all skills and topics. As seen in the table, our findings are consistent between GPT-4 and Claude-3 as graders (they have a high overlap), and the results corroborate our claim that GPT-4 is stricter than Claude-3.
>
> |                                      | k=2            | k=3            | k=4            | k=5            |
> |--------------------------------------|----------------|----------------|----------------|----------------|
> | Llama-13b-chat on SkillMix_train     | 0.24/0.31/0.19 | 0.02/0.07/0.01 | 0.01/0.06/0.01 | 0.00/0.00/0.00 |
> | -ft’ed on D(1,2,3) on Skillmix_train | 0.65/0.69/0.58 | 0.33/0.57/0.29 | 0.15/0.26/0.12 | 0.06/0.10/0.05 |

---

> ### Author Response · Authors · 2024-08-14
> **Did we address your questions or concerns?**
>
> Dear reviewer,
>
> Thank you again for your time and effort in reviewing our paper. As the discussion period deadline is approaching, we would like to know if our rebuttal addresses your questions or concerns and if there are any follow-up questions.
>
> Thank you

---

### Decision · Program_Chairs · 2024-09-25

**Decision:**

Accept (poster)

**Comment:**

This paper is borderline, but was positively received by the reviewers. Because the paper addresses an important topic (compositional generalization in LLMs) and demonstrates non-intuitive results (that generalization can be learned via fine-tuning), I believe the results will be interesting to the community and I therefore recommend acceptance.

That said, I encourage the authors to respond to the reviewers' concerns about the transparency of the evaluation protocol, and whether or not "cheating" is possible.